# When good for business is not good enough: Effects of pro-diversity beliefs and instrumentality of diversity on intergroup attitudes

Mathias Kauff[1]*, Katharina Schmid[2], Oliver Christ[3]

1 Department of Psychology, Medical School Hamburg, Hamburg, Germany, 2 ESADE Business School, Ramon Llull University, Barcelona, Spain, 3 Faculty of Psychology, FernUniversität in Hagen, Hagen, Germany

* mathias.kauff@medicalschool-hamburg.de

**Data Availability Statement:** All 4 files are available from the OSF database (https://osf.io/kyxvf).

## Abstract

Instrumentality-based pro-diversity beliefs (i.e., beliefs that diverse groups outperform homogenous groups in terms of group functioning) have been shown to improve intergroup attitudes. However, such valuing of diversity due to its expected instrumentality holds the risk that outgroups may be devalued in situations when diversity ends up being detrimental to group functioning. Across four experiments, we studied the interplay between instrumentality-based pro-diversity beliefs, actual instrumentality of ethnic diversity, and outgroup attitudes. Our results do not reveal a robust interaction effect between instrumentality-based pro-diversity beliefs and actual instrumentality of diverse groups. Some evidence, however, supports the assumption that instrumentality-based pro-diversity beliefs yielded a weaker positive or even a negative effect on ethnic outgroup attitudes when ethnic diversity was perceived as non-instrumental (i.e., when diversity was perceived as having a negative impact on group functioning). Theoretical contributions, practical implications, and directions for future research are discussed.

## Introduction

In recent years, diversity is often hailed as offering competitive and economic advantages to organizations and societies, a view that in organizational psychology has been described as a business case for diversity: diversity is valued because it is good for business [1]. Many organizations explicitly endorse pro-diversity statements that value diversity as an asset for their organizations. Google CEO Sundar Pichai, for example, states that "A diverse mix of voices leads to better discussions, decisions, and outcomes for everyone." [2]. In recent years, business case arguments can also be observed in debates about societal diversity. In an op-ed, political economist Will Hutton points to the value of ethnic diversity to societies by stating, that "it has been immigrants and refugees who have been part of the alchemy of any country's success" [3].

**Funding:** This research was supported by a grant from the Deutsche Forschungsgemeinschaft awarded to MK (KA4028/2-1). The funders had no role in study design, data collection and analysis, decision to publish, or preparation of the manuscript.

**Competing interests:** The authors have declared that no competing interests exist.

This idea, that diversity is valued because of its anticipated instrumentality for group functioning, is captured in the construct of (instrumentality-based) pro-diversity beliefs, which has received growing attention in recent research. Individuals holding such instrumentality-based pro-diversity beliefs value diversity because they believe that diversity is an asset for groups, that is diversity is expected to have a positive impact on group functioning. Several studies have pointed to desirable outcomes of such instrumentality-based pro-diversity beliefs, including positive intergroup relations, both on an organizational-level [4] as well as on a societal-level [5]. Noteworthy, however, is that such instrumentality-based pro-diversity beliefs underlie the assumption that diversity *is* instrumental for the functioning of groups; in other words, instrumentality-based pro-diversity beliefs yield positive outcomes *because of* the anticipated instrumentality of diversity. What has not yet been studied is the interplay between instrumentality-based pro-diversity beliefs and actual instrumentality of diversity (i.e., the consequences of cooperating within diverse groups, or the extent to which diversity in the group actually ends up benefitting group performance).

We argue that a business case for diversity, as reflected in instrumentality-based pro-diversity beliefs, may provide a rationale for devaluation of minority group members if the encountered diversity is not instrumental [6]. In other words, if diversity is not valued in itself but only because of its anticipated instrumental value for the functioning of groups, instrumentality-based pro-diversity beliefs may fail to positively impact intergroup attitudes or may even damage intergroup relations whenever positive expectations are not fulfilled. Thus, someone holding instrumentality-based pro-diversity beliefs may come to hold negative attitudes toward diverse outgroup members when the encountered diversity results in detrimental (as opposed to instrumental) group performance. To date, no empirical evidence exists to answer this question, as to how individuals holding or being presented with instrumentality-based pro-diversity beliefs react if diversity within a group ends up being non-instrumental for the group's performance and/or success.

In four studies situated in the context of ethnic diversity, we study the interplay between instrumentality-based pro-diversity beliefs and actual instrumentality of group diversity in shaping intergroup attitudes. We propose that the effect of instrumentality-based pro-diversity beliefs on outgroup attitudes is moderated by the actual instrumentality of diversity, to the extent that previously shown positive effects of instrumentality-based pro-diversity beliefs on outgroup attitudes should primarily occur when outgroup members are perceived as instrumental. Group diversity that is perceived as non-instrumental, or even as detrimental, on the other hand should lead to a deterioration of outgroup attitudes for individuals holding or being presented with instrumentality-based pro-diversity beliefs.

In order to test our assumptions, we measure instrumentality-based pro-diversity beliefs in Study 1 and experimentally contrast instrumentality-based pro-diversity beliefs with other, instrumentality-independent forms of pro-diversity beliefs (i.e. justice-based pro-diversity beliefs) in Studies 2–4.

## Pro-diversity beliefs and intergroup relations

Rooted in organizational psychology, instrumentality-based diversity beliefs are defined as beliefs about the optimal composition of work groups, such that individuals holding pro-diversity beliefs consider diverse groups to outperform homogenous groups [7]. Instrumentality-based diversity beliefs have been suggested as a moderating variable in the equivocal relationship between work group diversity and performance, where both positive and negative effects of diversity have been observed [8]. Research has, for example, shown that diversity can, on the one hand, foster creativity [9], but, on the other, stifle group performance [10]. Negative

consequences of diversity are partly attributable to the fact that diverse groups often consist of subgroups which can give rise to biased intergroup perceptions [8].

Van Knippenberg et al. [8] propose that instrumentality-based pro-diversity beliefs lead to a positive association between work group diversity and performance. Instrumentality-based pro-diversity beliefs are thought to reduce identity threat surrounding subgroup memberships and increase positive evaluations of subgroups, thereby facilitating creative task solutions [8]. In other words, the potential of diverse teams may best be exploited if team members hold instrumentality-based pro-diversity beliefs (i.e., if they value diversity because of its anticipated instrumentality). Confirming this view, Homan, van Knippenberg, Van Kleef and De Dreu [11] found that informationally diverse groups engaged in deeper information elaboration and performed better in a problem-solving task than homogenous groups when members were made to believe that diverse teams outperform homogenous ones, i.e., when they held instrumentality-based pro-diversity beliefs [12]. Furthermore, instrumentality-based pro-diversity beliefs have been found to increase identification with diverse groups [4].

In recent years, research on instrumentality-based pro-diversity beliefs has moved beyond the organizational realm, to study predictors and consequences of beliefs about the value of diversity at a more general, societal level, focusing in particular on the consequences of diversity beliefs for intergroup relations outcomes [13]. According to Kauff, Stegmann, van Dick, Beierlein, and Christ [14] instrumentality-based pro-diversity beliefs focusing on ethnic diversity on a societal level can be characterized as "beliefs that the society as a group can profit from ethnic and cultural diversity in achieving goals and solving tasks and problems" (p. 497). To date, research considering the consequences of instrumentality-based pro-diversity beliefs for intergroup attitudes has generally observed positive outcomes. In a field-experiment, Kauff, Issmer, and Nau [15, Study 2], for example, demonstrated that participants selected more immigrant candidates for a communal commission after reading an article emphasizing the superiority of ethnically diverse groups compared to participants that read unrelated control articles. Relatedly, Guerra, Gaertner, António, and Deegan [16] demonstrated that immigrant outgroups were more positively evaluated if they were seen as contributing to the functioning of society.

## Pro-diversity beliefs and the business case for diversity

Notwithstanding aforementioned findings on the positive consequences of instrumentality-based pro-diversity beliefs for intergroup relations, a critical aspect that needs to be kept in mind is that pro-diversity beliefs reflect beliefs concerning the *instrumentality* of diversity. The concept of instrumentality-based pro-diversity beliefs is thus much in line with the business case for diversity. Since the emergence of the business-case rationale in the 1980s, diversity has been deemed as useful for increasing businesses' productivity, improving organizational learning through new perspectives, and facilitating creative solutions for tasks [1]. In recent years, the business case for diversity has also been extended more widely, to debates about the value of diversity to societies at large. For example, in an executive order promoting diversity in the federal workforce, US-president Obama argued that diversity enables the USA to manage new challenges by considering diverse perspectives [17].

However, this view has also been subjected to criticism. Noon [6], for example, has argued that the "overly rational cost-benefit analysis" [6; p.778] of the business case for diversity poses the risk that low-status groups (e.g., ethnic minorities) may be negatively evaluated. Accordingly, Tomlinson and Schwabenland [18] argued that adopting a business case may justify devaluation and discrimination of minority group members if diversity is not instrumental.

These criticisms also apply to instrumentality-based pro-diversity beliefs, since they, at their core, underlie a business case for diversity: Diversity is only valued *because* of its expected

benefits, to society, organizations, work groups, etc. Yet as Kauff and Wagner [5] argue, the construct of pro-diversity beliefs "contains the danger that [they] can also legitimize the devaluation and discrimination of those minority groups which seemingly do not contribute to the economic and cultural prosperity of a relevant in-group" (p. 719). Hence, instrumentality-based pro-diversity beliefs may *only* lead to an improvement of intergroup attitudes as long as diversity is actually perceived as instrumental. Indeed, based on the reasoning by Noon [6] and others [1], instrumentality-based pro-diversity beliefs may legitimize the devaluation of outgroup members, that is it may even negatively impact intergroup attitudes if, counter to expectations, diversity ends up hindering group success. Besides being a legitimizing process, it might also be that instrumentality-based pro-diversity beliefs intensify subgroup-categorization which could also facilitate the devaluation of outgroup members–especially when expectations are not met [12].

The effect of pro-diversity beliefs that do not presuppose instrumentality of diverse groups and are based on justice-based considerations, however, should be unaffected by the instrumentality of the instrumentality of diversity. Justice-based pro-diversity beliefs do not build on an instrumentality-based business case for diversity. They do not imply the idea that diversity should be valued because of its anticipated instrumentality. Rather they refer to groups' and individuals' moral obligations to reduce inequality and injustice between social groups. In other words, in a justice-based pro-diversity beliefs framework diversity is valued because of individuals' duty to work against inequality and to support others that are in need [1, 6].

## Overview of the present research

In the present research, we study the interplay between instrumentality-based pro-diversity beliefs, actual instrumentality of diversity in groups (i.e., whether group functioning is positively or negatively affected by diversity) and outgroup attitudes. Specifically, we aim to answer the question of how individuals holding instrumentality-based pro-diversity beliefs (i.e. individuals valuing diversity because of its anticipated instrumentality) evaluate outgroups if the diversity in the group does not benefit group performance and/or success. We hypothesize the effect of instrumentality-based pro-diversity beliefs on attitudes towards outgroups to be moderated by the actual instrumentality of diverse groups. In line with previous research, we expect a positive effect of instrumentality-based pro-diversity beliefs on outgroup attitudes, but only if group diversity is instrumental. We do not expect a positive effect when diversity is detrimental–in fact there might even be a negative effect.

We tested our assumptions in four experimental studies situated in the context of ethnic diversity, focusing on a range of dependent variables tapping intergroup attitudes. While Study 1 focusses on ethnic diversity in the context of small-group collaboration in an online task, Studies 2 and 3 address ethnic diversity in society at large and Study 4 in the context of larger societal groups (i.e. universities). In Study 1, we measured participants' instrumentality-based pro-diversity beliefs. Participants then ostensibly collaborated on a task in a diverse group and were confronted with experimentally varied feedback about the outcome of these interactions (i.e., the actual instrumentality of diversity). In Studies 2–4, we aimed to address the causal effects of the instrumental core of pro-diversity beliefs. We here differentiate between instrumentality-based and justice-based (Studies 2–4) pro-diversity beliefs. In other words, in addition to manipulating instrumentality we manipulate the type of pro-diversity beliefs (instrumentality-based vs. justice-based pro-diversity beliefs). We used justice-based pro-diversity beliefs as control conditions because they imply a non-instrumental appreciation of diversity.

## Study 1

Study 1 was situated in the context of intergroup relations between non-immigrant Germans and immigrants. Participants were Germans without migration background who ostensibly collaborated with ethnic outgroup members. We first measured participants' instrumentality-based pro-diversity beliefs and then manipulated actual instrumentality by experimentally varying the outcome of a simulated cooperative task within ethnically diverse groups. Finally, intergroup attitudes were measured with items tapping positive feelings, warmth, and competence toward ethnic outgroup members.

## Method

All studies confirm to the Declaration of Helsinki. Studies were approved by the local ethics committee of the Psychology Department at the University of Marburg (Germany, 2013-24k). In all studies, written informed consent had to be given online on the first page, that is participants had to actively agree to take part in the study after having received information about the study, data handling, and risks of participation. Participants were aware that they could withdraw at any time without consequences. Data were completely anonymized before data storage. Written debriefings were given on the final page. We used clear and easy language to inform participants about the deceptive elements of the studies. Moreover, participants had the opportunity to contact the corresponding author should they have any questions. There is no conflict of interest to declare. Data of all studies as well as the application for ethical approval (in German) and the ethics statement (in German) are available from the Open Science Framework (https://osf.io/kyxvf). Please note, that in contrast to the planned research project outlined in the application for ethical approval, we only realized three instead of six studies and slightly modified some studies with regard to measurement or circumstantial research design matters. Moreover, we ran an additional replication study. Our procedure with regard to research ethics was not altered, however. Data of the four studies presented in this paper are exclusively used for these studies, that is the data reported here have not been, and will not be, used for other projects.

**Pretest.**   Prior to running Study 1, we pre-tested the experimental design in the context of intergroup relations between different university student groups at a German university ($n$ = 78; S1 File). Results of this pretest indicated that the effect of instrumentality-based beliefs about educational diversity on intergroup attitudes was moderated by the perceived instrumentality of diversity. More precisely, supposedly detrimental dyadic interactions between psychology students and economy students led to a deterioration of attitudes towards economics students among psychology students holding instrumentality-based educational pro-diversity beliefs. Because to our best knowledge no other studies have addressed an interaction between instrumentality-based pro-diversity beliefs and perceived instrumentality of diversity we used the effect size found in the pretest as a basis for power calculations (although the design of Study 1 differed in some regards from the pretest design). Based on an effect size of $\Delta R^2$ = .119, $\alpha$ = .05 and a power (1-$\beta$) of .80 we aimed for a sample of around 100 participants for Study 1.

**Overview.**   Study 1 was conducted online. Participants ostensibly collaborated with other participants within a "virtual group" on a brainstorming task, that is they were told that their responses were combined with responses from previous participants. Participants were randomly assigned to one of four conditions: 1. instrumental diverse condition, 2. neutral diverse condition, 3. detrimental diverse condition, and 4. detrimental non-diverse condition. Instrumentality was manipulated by providing fictitious feedback about the group performance in a brainstorming task. In conditions 1 to 3, participants either received feedback indicating that

results of their diverse group (Germans and immigrants/Germans with a migration background) were above average (instrumental diverse), average (neutral diverse), or below average (detrimental diverse) compared to past results of non-diverse teams. In condition 4 (detrimental non-diverse) participants worked within a homogenous group (Germans only) and received feedback that the group's results were below average compared to prior teams' results. We considered it important to include this negative feedback condition involving a non-diverse group to allow for comparison of effects between this condition and the condition involving a detrimental collaboration in a diverse group. Effects of the detrimental non-diverse condition on the relationship between instrumentality-based pro-diversity beliefs and out-group attitudes would suggest that a detrimental collaboration has an effect–independent of the composition of a group. In other words, by comparing effects of the detrimental diverse condition with the detrimental non-diverse condition we can rule out a mere effect of negative feedback.

**Participants.** Data was collected between June and July 2015. In total 136 participants completed the online-experiment. Four participants were excluded because they spent less than ten minutes completing the questionnaire (criteria based on duration of test runs; mean time spent completing the questionnaire was 22 minutes, *SD* = 12 minutes). We tested our assumptions in a sample of non-immigrant participants to ensure that the diverse composition of teams was perceived similarly for all participants. We, hence, excluded an additional 23 participants because they had a nationality or mother tongue other than German. Of the remaining 109 participants (mean age = 34.0, *SD* = 10.0), 74 were women and 35 men. Eighty-one participants were psychology students who participated in return for course credit. The remaining participants were recruited in Facebook groups. They received a lot for a raffle in return for participation. The number of participants per condition ranged from 20 to 36.

**Procedure.** We employed extensive measures to disguise the overall research question and reduce demand characteristics. For example, participants were told that they would be taking part in three independent studies on social issues, virtual cooperation, and attitudes towards different social groups. Accordingly, we included several distractor items. Moreover, participants were told that the order of the studies was randomized and that some demographic questions would be repeated within each study. Accordingly, responses to an open question asking for comments about the study did not indicate that participants had guessed the research question.

All participants first answered questions related to different social issues, among them their attitudes towards ethnic diversity (i.e. instrumentality-based pro-diversity beliefs). In the next part, participants were told that they had to work on a task related to marketing and that their answers would be used to build a pool of answers from different participants. In the task, participants had to choose ten words from a list of 51 words that could be used for an advertisement. Participants had to complete two trials, selecting words for a fair-trade smartphone as well as for sneakers that imitate the feeling of running barefoot. In order to simulate cooperation in a virtual-group participants were further told that their answers would be randomly combined with answers from three previous respondents. They were also told that they would receive feedback on the result of this collaboration: A group score would be calculated on the basis of the quality of the selected words, based on expert ratings. Moreover, participants were told to bring in their individual perspective and to try to create unique solutions, to avoid words being selected by more than one participant in the virtual-group. Participants then completed a short questionnaire including demographic variables (e.g., nationality) and personality items. Finally, after having selected ten words, participants received feedback about the virtual-group's result. Table 1 provides an example for how feedback was presented to participants. The feedback included information on the virtual cooperation partners, that is their

**Table 1. Example feedback for a 22-year-old male participant in the detrimental diverse group condition of Study 1 (second trial).**

| No. | Nationality | Age | Gender | N | E | O | A | C |
| --- | --- | --- | --- | --- | --- | --- | --- | --- |
| 12 | Moroccan | 24 | Male | 0 | + | 0 | 0 | + |
| 145 | German | 21 | Male | 0 | 0 | - | + | 0 |
| 037 | German/Turkish | 25 | Male | 0 | + | 0 | 0 | 0 |

(0 = average, + slightly above average, ++ strongly above average,—slightly below average, strongly below average; N stands for Neuroticism. Individuals with high scores in N are emotional and vulnerable. E stands for Extraversion. Individuals with high scores in E are sociable and not at all reserved. O stands for Openness. Individuals with high scores in O are curious and innovative. A stands for Agreeableness. Individuals with high scores in A are friendly and sympathetic. C stands for Conscientiousness. Individuals with high scores in C are well-organized and concerned.

Group feedback: Your group obtained **28** points. This result is **below average**. (Information: The result of your virtual group has been compared to results of groups in a previous study.)

Nationality and group feedback was experimentally varied across conditions. Values for personality variables were held constant across conditions. Age and gender were matched to participants' information (i.e. age was calculated as participants' age +2, -1 and +3 respectively; gender was the same as participants').

nationality, age, gender, and personality variables. Information on personality variables were held constant across conditions, while gender and age were matched to participants' information on these respective variables (i.e., same gender; age within a range of participant's age -2 to +4) to reduce diversity on these dimensions. Importantly, as mentioned above, we varied cooperation partners' ethnicity across conditions to manipulate ethnic diversity of virtual-groups: In the detrimental non-diverse condition all cooperation partners were German. In the instrumental diverse, neutral diverse, and detrimental diverse condition collaboration partners were German, German, and Turkish (first trial) and Moroccan, German, and German-Turkish (second trial). The feedback also included information on the virtual-group's performance in comparison to past results. Feedback was experimentally varied across conditions to manipulate the instrumentality of the collaboration: In the instrumental diverse condition participants read that their group result was above average (93 points in the first and 85 points in the second trial). In the neutral diverse condition participants read that their group result was average (61/58 points). In the detrimental diverse as well as the detrimental non-diverse conditions participants read that their group result was below average (35/28 points). We refrained from including a manipulation check because we did not want to endanger the plausibility of our cover story (i.e., that participants were taking part in three independent studies). However, results of the pretest indicate that a comparable procedure successfully altered perceptions of instrumentality.

After completion of the task, participants were redirected to the next part of the study, the ostensible third study. Participants were told that the purpose of this part of the study was simply to collect data from a reference sample for another study, that is their answers would be used as a comparison group for a sample of juvenile criminals that was supposedly surveyed in the context of another study. We assumed that for participants familiar with our departments' research focus a study on juvenile criminals would constitute a believable research project.

After first answering a number of demographic questions, participants were asked to evaluate different social groups. Participants then rated their attitudes as well as perceptions of warmth and competence towards different social groups (i.e., immigrants, politicians, unemployed people, wealthy people, and homosexuals). Upon completing the questionnaire participants were thanked and debriefed.

**Measures.** Unless otherwise indicated, all items were answered on 5-point-scales ranging from 1 = *do not agree at all* to 5 = *totally agree*. If constructs were measured with more than one item responses to these items were aggregated to form a composite score. *Instrumentality-based pro-diversity beliefs* were measured with seven items (e.g., 'I value ethnic diversity

because it benefits the country.', 'Culturally diverse groups are usually more productive than culturally homogenous groups.'; α = .914).

Attitudes towards immigrants were measured with one feeling thermometer item ('In general, how would you evaluate immigrants?'; scaling from 1 = *very negative* to 10 = *very positive*; [19]) measuring *general attitudes towards immigrants*, as well as additional items measuring immigrants' warmth and competence [20]. *Warmth* was measured with three items ('friendly', 'warm', 'likeable'; α = .878). *Competence* was also supposed to be measured with three items. However, we excluded one item ('dependent') because it decreased the scale's reliability (remaining items: 'competent', 'efficient'; *r* = .848, *p* < .001; reliability prior to exclusion of item: α = .505).

We also included items measuring general attitudes, warmth and competence targeting four additional outgroups (i.e., politicians, unemployed people, wealthy people, and homosexuals; S2 File). In addition, a number of unrelated distractor items were included (i.e., items focusing on climate-change, demographic change, the financial status of the EU, and made-up personality questionnaire items, as well as one item measuring political orientation).

Political orientation was measured prior to the manipulation as a potential covariate [21]. In accordance with Wang, Sparks, Gonzales, Hess, & Ledgerwood [22], we had planned to include political orientation in our models only in case of high correlations with the dependent variables. Since the correlations between political orientation and dependent variables in Study 1, were only small to moderate however (general attitudes: *r* = -.368, *p* < .001; warmth: *r* = -.314, *p* = .001; competence: *r* = -.193, *p* = .044), we refrained from including political orientation as a covariate. Nevertheless, results with inclusion of this covariate can be found in the Supporting Material (S1 Table). They were comparable with the results obtained without the covariate political orientation.

## Results and discussion

Table 2 lists the descriptive statistics as well as intercorrelations between the measures. Please note, that we additionally ran analyses prior to exclusion of participants that had a nationality or mother tongue other than German. Results were comparable with the results for the reduced sample (see S2 Table for a detailed description of results).

We tested whether the effect of instrumentality-based pro-diversity beliefs on attitudes towards immigrants (i.e. general attitudes, warmth and competence) was moderated by instrumentality in diverse groups (dummy coded with three variables: a) instrumental diverse as a baseline condition vs. neutral diverse, b) instrumental diverse vs. detrimental diverse, and c) instrumental diverse vs. detrimental non-diverse). Dummy-coding in regression analyses with a multicategorical moderator allows for testing the moderation effect of each category (here: experimental condition) against a preassigned reference group [e.g., 23]. We assume that previously found prejudice-reducing effects of instrumentality-based pro-diversity beliefs

**Table 2. Means, standard deviations, and intercorrelations of measures of Study 1.**

|  | *M* | *SD* | **2** | **3** | **4** |
|---|---|---|---|---|---|
| 1 pro-diversity beliefs | 3.79 | 0.82 | .66** | .49** | .47** |
| 2 general attitudes | 6.31 | 1.89 |  | .63** | .63** |
| 3 warmth | 3.31 | 0.68 |  |  | .72** |
| 4 competence | 3.23 | 0.73 |  |  |  |

*N* = 109; * *p* < .05

** *p* < .01, *** *p* < .001

**Table 3. Results of moderated regression analyses of Study 1.**

| | general attitudes | | | | warmth | | | | competence | | | |
|---|---|---|---|---|---|---|---|---|---|---|---|---|
| | $b$ | SE | $p$ | $CI_{95\%}$ | $b$ | SE | $p$ | $CI_{95\%}$ | $b$ | SE | $p$ | $CI_{95\%}$ |
| constant | -1.497 | 1.172 | .205 | -3.823, 0.829 | 1.120 | 0.481 | .022 | 0.166, 2.074 | 1.014 | 0.515 | .051 | -0.006, 2.035 |
| pro-diversity beliefs | 2.107 | 0.302 | .001 | 1.508, 2.706 | 0.606 | 0.125 | .001 | 0.359, 0.853 | 0.626 | 0.133 | .001 | 0.361, 0.890 |
| neutral div. vs. instr. div. (D1) | 2.309 | 1.927 | .234 | -1.514, 6.131 | 0.450 | 0.801 | .575 | -1.138, 2.038 | 0.855 | 0.857 | .320 | -0.844, 2.555 |
| detrimental div. vs. instr. div. (D2) | 3.434 | 1.558 | .030 | 0.342, 6.526 | 1.163 | 0.638 | .071 | -0.103, 2.428 | 1.249 | 0.683 | .070 | -0.106, 2.603 |
| detrimental non-div. vs. instr. div. (D3) | 2.243 | 2.372 | .347 | -2.465, 6,950 | 0.781 | 0.988 | .431 | -1.179, 2.742 | -0.766 | 1.057 | .470 | -2.864, 1.332 |
| D1 X pro-diversity beliefs | -0.610 | 0.487 | .213 | -1.576, 0.355 | -0.150 | 0.203 | .460 | -0.552, 0.252 | -0.299 | 0.217 | .171 | -0.729, 0.131 |
| D2 X pro-diversity beliefs | -0.956 | 0.405 | .020 | -1.759, -0.153 | -0.328 | 0.167 | .052 | -0.659, 0.003 | -0.349 | 0.179 | .053 | -0.704, 0.005 |
| D3 X pro-diversity beliefs | -0.800 | 0.607 | .191 | -2.005, 0.405 | -0.288 | 0.253 | .259 | -0.790, 0.215 | 0.099 | 0.271 | .716 | -0.439, 0.637 |
| $R^2$ | $R^2 = .486, F(7, 99) = 13.359, p = .001$ | | | | $R^2 = .298, F(7, 101) = 6.125, p = .001$ | | | | $R^2 = .304, F(7, 101) = 6.298, p = .001$ | | | |
| $R^2$ change due to interaction | $\Delta R^2 = .030., F(3, 99) = 1.940, p = .128$ | | | | $\Delta R^2 = .029, F(3, 101) = 1.375, p = .255$ | | | | $\Delta R^2 = .040, F(3, 101) = 1.912, p = .132$ | | | |

instr. = instrumental, div. = diversity

occurred because diversity was perceived as instrumental. Building on this assumption, we used the instrumental diverse condition as a reference group to contrast the effects on non-instrumentality (neutral diverse condition) and negative instrumentality (detrimental diverse and detrimental non-diverse conditions) with positive instrumentality (instrumental diverse).

In this section, we focus on the interaction effects. Full results are, however, depicted in Table 3. Testing the detrimental diverse condition against the instrumental diverse condition, we observed a significant interaction effect of negative instrumentality on the relationship between instrumentality-based pro-diversity beliefs and general outgroup attitudes ($b = -0.956$, $SE = 0.405$, $p = .020$, $CI_{95\%} = -1.759, -0.153$). In other words, detrimental diversity (compared to instrumental diversity) had a mitigating effect on the positive relation between instrumentality-based pro-diversity beliefs and general outgroup attitudes. Neither non-instrumentality in diverse (i.e., neutral diverse vs. instrumental diverse condition; $b = -0.610$, $SE = 0.487$, $p = .213$, $CI_{95\%} = -1.576, .355$) nor negative instrumentality in non-diverse groups (i.e., detrimental non-diverse vs. instrumental diverse condition; $b = -0.800$, $SE = 0.607$, $p = .191$, $CI_{95\%} = -2.005, 0.405$) interacted with instrumentality-based pro-diversity beliefs. In other words, neither a neutral outcome (non-instrumental but not detrimental) in a diverse group nor a negative detrimental outcome in a non-diverse condition had an effect on instrumentality-based pro-diversity beliefs and general outgroup attitudes.

Analyses of conditional effects revealed that instrumentality-based pro-diversity beliefs had a significant positive effect on general attitudes in all conditions (instrumental diverse: $b = 2.107$, $SE = 0.302$, $p < .001$, $CI_{95\%} = 1.508, 2.706$; neutral diverse: $b = 1.497$, $SE = 0.382$, $p = .002$, $CI_{95\%} = 0.739, 2.254$; detrimental diverse: $b = 1.151$, $SE = 0.270$, $p < .001$, $CI_{95\%} = 0.616, 1.686$; detrimental non-diverse: $b = 1.307$, $SE = 0.527$, $p = .015$, $CI_{95\%} = 0.261, 2.352$). Graphical inspection of the conditional effects (Fig 1) suggests that the effect of instrumentality-based pro-diversity beliefs was smallest in the detrimental diverse condition. To test whether the interaction effect of non-instrumentality on the relationship between instrumentality-based pro-diversity beliefs and general outgroup attitudes was specific to interactions within a diverse group we additionally contrasted the detrimental diverse condition with the detrimental non-diverse condition as a moderator in a moderated regression. A contrast of both conditions did not significantly moderate the effect of instrumentality-based pro-diversity beliefs on general outgroup attitudes ($b = -.156$, $SE = 0.666$, $p = .816$, $CI_{95\%} = -1.492, 1.180$)–the effect of instrumentality-based pro-diversity beliefs on general outgroup attitudes did not differ in

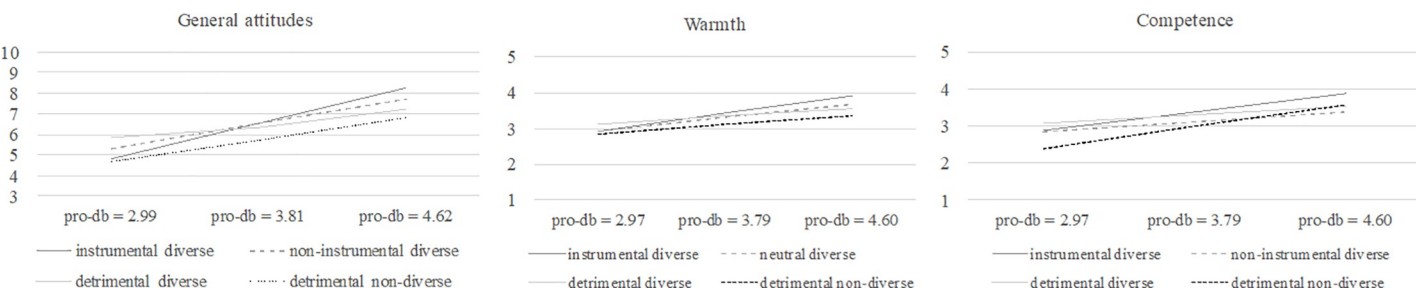

**Fig 1. Effects of pro-diversity beliefs and instrumentality on general attitudes, warmth, and competence in Study 1.** pro-db = pro-diversity beliefs.

dependence of whether a detrimental interaction occurred in a diverse or a homogenous group.

Concerning the dependent variable warmth, we observed a marginally significant interaction effect of negative instrumentality (i.e., detrimental diverse vs. instrumental diverse condition) on the relationship between instrumentality-based pro-diversity beliefs and warmth ($b = -0.328$, $SE = 0.167$, $p = .052$, $CI_{95\%} = -0.659, 0.003$). In other words, detrimental diversity (compared to instrumental diversity) had a marginally significant mitigating effect on the positive relation between instrumentality-based pro-diversity beliefs and warmth. Neither non-instrumentality in diverse groups (neutral diverse vs. instrumental diverse condition; $b = -0.150$, $SE = 0.203$, $p = .460$, $CI_{95\%} = -0.552, 0.252$) nor negative instrumentality in non-diverse groups (detrimental non-diverse vs. instrumental diverse condition; $b = -0.288$, $SE = 0.253$, $p = .259$, $CI_{95\%} = -0.790, 0.215$) interacted with instrumentality-based pro-diversity beliefs. Instrumentality-based pro-diversity beliefs had a significant positive effect on warmth in all conditions involving collaboration in diverse groups (instrumental diverse: $b = 0.606$, $SE = 0.125$, $p < .001$, $CI_{95\%} = 0.359, 0.853$; neutral diverse: $b = 0.56$, $SE = 0.160$, $p = .005$, $CI_{95\%} = 0.139, 0.773$; detrimental diverse: $b = 0.278$, $SE = 0.111$, $p = .014$, $CI_{95\%} = 0.058, 0.498$) but not in the detrimental non-diverse condition ($b = 0.319$, $SE = 0.221$, $p = .152$, $CI_{95\%} = -0.119, 0.756$; Fig 1). Results of an additional analysis indicate that the contrast between the detrimental diverse condition and the detrimental non-diverse condition did not moderate the effect of instrumentality-based pro-diversity beliefs on warmth ($b = -.040$, $SE = 0.275$, $p = .884$, $CI_{95\%} = -0.591, 0.511$). As for general outgroup attitudes, the effect of instrumentality-based pro-diversity beliefs on warmth does not differ in dependence of whether a detrimental interaction occurred in a diverse or a homogenous group. The pattern for competence was comparable to the pattern for general attitudes and warmth (interaction effect negative instrumentality (detrimental diverse vs. instrumental diverse condition) X instrumentality-based pro-diversity beliefs: $b = -0.349$, $SE = 0.179$, $p = .053$, $CI_{95\%} = -0.704, 0.005$; interaction effect non-instrumentality (neutral diverse vs. instrumental diverse condition) X instrumentality-based pro-diversity beliefs: $b = -0.299$, $SE = 0.217$, $p = .171$, $CI_{95\%} = -0.729, 0.131$; negative instrumentality in non-diverse groups (detrimental non-diverse vs. instrumental diverse condition) X instrumentality-based pro-diversity beliefs: $b = 0.099$, $SE = 0.271$, $p = .716$, $CI_{95\%} = -0.439, 0.637$). Instrumentality-based pro-diversity beliefs had a (marginally) significant positive effect on competence scores in all conditions (instrumental diverse: $b = 0.626$, $SE = 0.133$, $p < .001$, $CI_{95\%} = 0.361, 0.890$; neutral diverse: $b = 0.326$, $SE = 0.171$, $p = .059$, $CI_{95\%} = -0.013, 0.665$; detrimental diverse: $b = 0.276$, $SE = 0.119$, $p = .022$, $CI_{95\%} = 0.041, 0.512$; detrimental non-diverse: $b = 0.724$, $SE = 0.236$, $p = .003$, $CI_{95\%} = 0.256, 1.193$; Fig 1).

Results of an additional analysis revealed a marginally significant moderating effect of the contrast between the detrimental diverse condition and the detrimental non-diverse condition on the effect of instrumentality-based pro-diversity beliefs on competence ($b = -.448$,

$SE$ = 0.260, $p$ = .09, $CI_{95\%}$ = -0.969, 0.073). For competence, we see a trend indicating that the effect of instrumentality-based pro-diversity beliefs was smaller after a detrimental interaction within a diverse group than after a detrimental interaction within a homogenous group (conditional effects: detrimental diverse condition: $b$ = .276, $SE$ = 0.117, $p$ = .022, $CI_{95\%}$ = 0.420, 0.510; detrimental non-diverse condition: $b$ = -.724, $SE$ = 0.232, $p$ = .003, $CI_{95\%}$ = 0.259, 1.190).

Results of Study 1, at best, weakly support our assumptions. Contrary to theorizing [6], we did not find a *negative* effect of instrumentality-based pro-diversity beliefs on favorable outgroup attitudes after non-instrumental (i.e., neutral) or detrimental collaborations. However, we did obtain some weak evidence for a mitigating effect of participants' collaboration in diverse groups that resulted in detrimental group performance on the positive relationship between instrumentality-based pro-diversity beliefs and favorable outgroup attitudes. However, this evidence is somewhat compromised by the fact that we only found a marginally significant interaction effect when comparing the influence of detrimental interactions within diverse vs. non-diverse groups for competence–but not for general outgroup attitudes and warmth. Accordingly, we cannot rule out a mere effect of negative feedback.

In sum, Study 1 provided only weak support for the notion that detrimental interactions in diverse groups influence the relationship between instrumentality-based pro-diversity beliefs and outgroup attitudes. This might be due to a relatively small sample size as well as the somewhat artificial context of virtual groups. We therefore sought to conduct additional studies (Studies 2–4) to include more ecologically valid contexts. Moreover, while we measured pro-diversity beliefs in Study 1 as a stable attitudinal construct, we extended our research in Studies 2–4 to manipulate different types of pro-diversity beliefs. Prior research suggests that instrumentality-based pro-diversity beliefs can (temporarily) be altered by experimental methods [5]. This change in the experimental design allowed us to further improve our research in two regards. First, we were able to introduce type of pro-diversity beliefs as an additional experimental factor, enabling us to more directly study causal processes. Second, we were able to directly isolate instrumentality-based valuing of diversity (i.e., pro-diversity beliefs underlying an instrumental orientation) from differently motivated valuing of diversity (i.e., pro-diversity beliefs underlying justice-based reasons (Studies 2–4)). For Studies 2–4, we hypothesized that differences in outgroup attitudes based on different levels of instrumentality of diversity (i.e. instrumental vs. detrimental) would only emerge when pro-diversity beliefs underlying an instrumental orientation (i.e., instrumentality-based pro-diversity beliefs) are experimentally activated but not when pro-diversity beliefs underlying different non-instrumental motivation are activated.

## Study 2

In Study 2, we tested our predictions in the context of intergroup relations between Germans and refugees, thus setting up our research against the backdrop of the so-called European 'refugee-crisis 2015'. Participants were non-immigrant German students. In addition to the experimental variation of instrumentality (instrumental vs. detrimental) we also manipulated type of pro-diversity beliefs (instrumentality-based vs. justice-based pro-diversity beliefs) by presenting participants with favorable views concerning diversity that stress its instrumental value (vs. views that stress its value on justice-based grounds). Finally, intergroup attitudes were measured with items tapping prejudice as well as social distance towards refugees.

### Method

**Participants.**  Data was collected in October 2015. Participants were psychology students who participated in return for course credit. We recruited as many participants as possible but

set a fixed closing date for the questionnaire. In total, 492 participants completed the questionnaire. We excluded 32 participants because they failed to correctly answer an attention check question and 45 participants because they spent less than ten seconds on one or both of the pages providing the texts that contained the manipulations (criterion based on duration of test runs; mean time spent on the page comprising the first manipulation was 54.78 seconds ($SD$ = 43.30), mean time spent on the page comprising the second manipulation was 46.07 seconds ($SD$ = 63.01)). Because we were interested in relationships among ethnic majority members an additional 109 individuals were excluded because they had a migration background. Of the remaining 306 participants (mean age = 32.57, $SD$ = 9.53), 235 were women, 68 men, and three indicated that they were unwilling to answer the question or identified with a different gender group. The number of participants per condition ranged from 65 to 86.

**Procedure.**   Study 2 was conducted online. It was announced as a survey on attitudes towards refugees and the refugee crisis in 2015. The study was conducted as a 2x2-design: We manipulated two types of pro-diversity beliefs (instrumentality-based vs. justice-based pro-diversity beliefs) and the actual instrumentality of refugees for communities (instrumental vs. detrimental).

Participants first answered a number of demographic questions, items on contact experiences with refugees and a single item asking for their political orientation. After that participants read two texts which they were told provided "basic information on the topic of the refugee crisis". The texts contained manipulations of two independent factors. In the first text, participants received information on how many refugees arrived in Europe, refugees' motivation for moving to Europe and their different countries of origin. Moreover, participants were confronted with a reason for why refugees should be supported in Germany: We either listed instrumentality-based or justice-based arguments (i.e., instrumentality-based vs. justice-based pro-diversity beliefs). Dependent on the conditions the respective paragraphs read:

> Experts and politicians point out that Germany can profit from supporting refugees. The past has shown that the German economy benefits from immigration and diversity.

> Experts and politicians point out that Germany has a moral obligation to support refugees. Germany has ratified the Universal Declaration of Human Rights and is hence bound to protect the rights to life, liberty, and security of person for every human.

Participants then read another text dealing with the consequences of refugee migration. This text was used to manipulate the actual instrumentality of refugees. Participants were given the example of a fictitious German village in which a refugee home had been introduced ten years earlier. In the instrumental condition they read that the introduction of the refugee home was a success story (e.g., in terms of refugees' work in small and medium-sized local businesses and the provision of foreign language courses for Germans provided by refugees). In the detrimental condition they read that it was "anything but a success story" (e.g., unfulfilled expectations of refugees not filling vacant positions in local businesses and lack of exchange between refugees and locals). In a next step, we measured two distractor items asking for participants' knowledge about the topic and whether the texts contained new information for them. After that, the dependent variables prejudice towards refugees and social distance were measured. Finally, participants answered manipulation check items and a number of additional distractor items asking them to rate how political institutions were dealing with the refugee crisis. The distractor items were presented on the same page as the items measuring the dependent variable. After having completed the questionnaire participants were thanked and debriefed.

**Table 4. Means, standard deviations, and intercorrelations of measures of Study 2.**

| | | | *M* | *SD* | 2 | 3 | 4 |
|---|---|---|---|---|---|---|---|
| 1 perceived instrumentality | justice-based pro-diversity beliefs | instrumental | 5.78 | 1.21 | -.35*** | -.34*** | -.06 |
| | | detrimental | 2.20 | 1.17 | | | |
| | instrumental pro-diversity beliefs | instrumental | 5.99 | 1.23 | | | |
| | | detrimental | 2.24 | 1.17 | | | |
| | general | | 4.08 | 2.19 | | | |
| 2 social distance | justice-based pro-diversity beliefs | instrumental | 3.15 | 1.46 | | .67*** | .47*** |
| | | detrimental | 3.49 | 1.52 | | | |
| | instrumental pro-diversity beliefs | instrumental | 2.96 | 1.47 | | | |
| | | detrimental | 3.13 | 1.45 | | | |
| | general | | 3.19 | 1.48 | | | |
| 3 prejudice tw. refugees | justice-based pro-diversity beliefs | instrumental | 3.25 | 1.59 | | | .53*** |
| | | detrimental | 3.57 | 1.58 | | | |
| | instrumental pro-diversity beliefs | instrumental | 2.97 | 1.47 | | | |
| | | detrimental | 3.17 | 1.62 | | | |
| | general | | 3.25 | 1.58 | | | |
| 4 political orientation | justice-based pro-diversity beliefs | instrumental | 3.16 | 1.14 | | | |
| | | detrimental | 3.17 | 1.12 | | | |
| | instrumental pro-diversity beliefs | instrumental | 3.45 | 1.29 | | | |
| | | detrimental | 2.91 | 1.18 | | | |
| | general | | 3.18 | 1.19 | | | |

*** $p < .001$

**Measures.** All items were answered on a 7-point-scale ranging from 1 = *do not agree at all* to 7 = *totally agree*. Items measuring the same construct were aggregated to form a composite score. *Prejudice towards refugees* was measured with four items (e.g., 'There are too many refugees living in Germany', 'Refugees are a burden for the social welfare system'; α = .876; [24]). *Social distance* was measured with four items (e.g., 'I would be willing to invite refugees to my home.' (reversed), 'I would be willing to visit refugees in a refugee home.' (reversed); α = .843; [25]). The manipulation check *perceived instrumentality* was measured with two items ('Citizens of Schwalmtal have profited from the refugee home.', 'The refugee home is a benefit to Schwalmtal.'; $r = .928$, $p < .001$).

Moreover, we measured *political orientation* as a potential covariate, alongside a number of unrelated items (i.e., items referring to the information texts, contact with refugees or items focusing on political reactions to the refugee crisis) that served as distractor items.

## Results and discussion

Table 4 lists the descriptive statistics as well as intercorrelations between the measures. Note that we additionally ran analyses prior to exclusion of participants with migration background. Results were comparable with the results for the reduced sample (S3 Table).

Correlations between political orientation and prejudice ($r = .533$, $p < .001$) as well as social distance ($r = .464$, $p < .001$) were large. Hence, we included political orientation as a covariate in the respective analyses [22]. As mentioned above, we also measured intergroup contact with refugees prior to the manipulation. Because contact was not correlated with neither prejudice ($r = .016$, $p = .766$) nor social distance ($r = .086$, $p = .133$) we refrained from including it as an additional covariate.

**Table 5.  Results of 2-factorial ANOVAs in Study 2.**

| | prejudice | | | | social distance | | | |
|---|---|---|---|---|---|---|---|---|
| | *F* | *df* | *p* | partial η² | *F* | *df* | *p* | partial η² |
| corrected model | 34.81 | 4 | .001 | .318 | 24.40 | 4 | .001 | .245 |
| constant | 18.30 | 1 | .001 | .058 | 35.74 | 1 | .001 | .106 |
| political orientation | 130.52 | 1 | .001 | .305 | 90.49 | 1 | .001 | .231 |
| pro-diversity beliefs (justice vs. instrumental) | 5.11 | 1 | .025 | .017 | 3.60 | 1 | .059 | .012 |
| instrumentality of refugees (instrumental vs. detrimental) | 8.25 | 1 | .004 | .027 | 7.34 | 1 | .007 | .024 |
| pro-diversity beliefs X instrumentality of refugees | 0.80 | 1 | .373 | .003 | 0.30 | 1 | .588 | .001 |
| error | | 298 | | | | 301 | | |
| *R²* | | | | .318 | | | | .245 |

We first tested whether our manipulation of instrumentality was successful. Perceived instrumentality should be influenced by the instrumentality manipulation but not by the type of pro-diversity beliefs manipulation. Results of a 2-factorial ANOVA supported these assumptions ($F_{(3, 301)} = 240.36$, $p < .001$, partial $\eta^2 = .706$; main effect of type of pro-diversity beliefs: $F_{(1, 301)} = 0.85$, $p = .36$, partial $\eta^2 = .003$; main effect of instrumentality: $F_{(1, 301)} = 710.45$, $p < .001$, partial $\eta^2 = .702$; interaction effect: $F_{(1, 301)} = 0.41$, $p = .52$, partial $\eta^2 = .001$). Perceived instrumentality was higher in the instrumental conditions ($M = 5.88$, $SD = 1.22$) than in the detrimental conditions ($M = 2.22$, $SD = 1.17$).

We then tested whether type of pro-diversity beliefs and instrumentality interacted: No significant interaction occurred (prejudice: $F_{(1, 298)} = 0.80$, $p = .373$, partial η2 = .003; social distance: $F_{(1, 301)} = 0.30$, $p = .588$, partial η2 = .001; full results are depicted in Table 5). Next, we ran simple main effects contrasting the factor instrumentality (instrumental vs. detrimental) with type of pro-diversity beliefs (i.e. instrumentality-based vs. justice-based). We observed a significant positive effect of the detrimental condition (compared to the instrumental condition) on prejudice in the instrumentality-based pro-diversity beliefs conditions ($F_{(1, 298)} = 6.58$, $p = .011$, partial η2 = .022) but no effect in the justice-based pro-diversity beliefs conditions ($F_{(1, 298)} = 2.12$, $p = .147$, partial η2 = .007). A similar pattern emerged for social distance: We found a positive effect of the detrimental condition (compared to the instrumental condition) on social distance in the instrumentality-based pro-diversity beliefs conditions ($F_{(1, 301)} = 4.89$, $p = .028$, partial η2 = .016) but no effect in the justice-based pro-diversity beliefs conditions ($F_{(1, 301)} = 2.55$, $p = .112$, partial η2 = .008). In other words, analyses of simple main effects suggest that an effect of instrumentality on prejudice and social distance only occurred when instrumentality-based pro-diversity beliefs were activated (but not when justice-based pro-diversity beliefs were salient). Note, however, that both detrimental conditions do not significantly differ from each other with regard to prejudice or social distance (prejudice: $F_{(1, 147)} = 0.97$, $p = .327$, partial η2 = .007; social distance: $F_{(1, 147)} = 0.97$, $p = .327$, partial η2 = .007; controlled for covariate political orientation). Fig 2 illustrates the results.

To sum up, in line with our assumptions, compared to justice-based pro-diversity beliefs (which had a negative effect on attitudes towards refugees, see discussion below) instrumentality-based pro-diversity beliefs positively influenced attitudes–but only when the presence of refugees in the town was described as successful, i.e., as instrumental. Importantly, no difference between the type of pro-diversity beliefs emerged for outgroup attitudes when the presence of refugees was framed as detrimental. In other words and contrary to our assumptions, we did not observe a negative effect of negative instrumentality on outgroup attitudes when instrumentality-based pro-diversity beliefs (compared to justice-based pro-diversity beliefs)

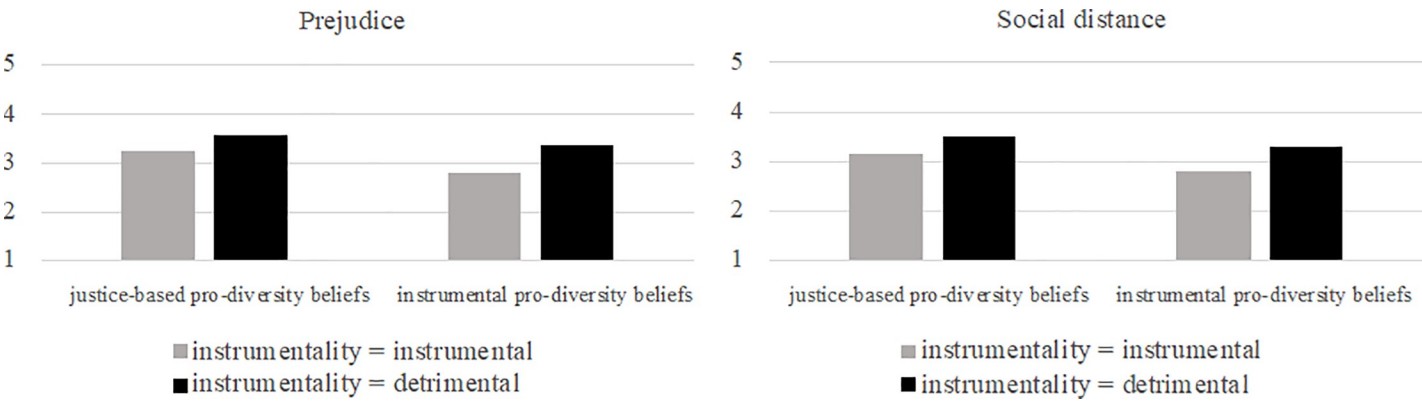

**Fig 2. Effects of pro-diversity beliefs (instrumentality- vs. justice-based pro-diversity beliefs) and instrumentality (instrumental vs. detrimental) on prejudice, and social distance in Study 2.** Covariate political orientation is evaluated at 3.18.

were made salient. On the contrary, however, a positive effect of instrumentality-based pro-diversity beliefs was dependent on the salience of positive instrumentality.

It is noteworthy that results differ when the covariate political orientation is excluded: Results for analyses without political orientation show a similar pattern. However, results do not meet the threshold for statistical significance (S4 Table). We neither observed an effect of the detrimental condition (compared to the instrumental condition) on prejudice in the instrumentality-based pro-diversity beliefs condition (F(1, 299) = 1.61, p = .21, $\eta^2$ = .005) nor in the justice-based pro-diversity beliefs condition (F(1, 299) = 0.53, p = .47, $\eta^2$ = .002). More-over, we neither observed an effect of the detrimental condition (compared to the instrumental condition) on social distance in the instrumentality-based pro-diversity beliefs condition (F(1, 299) = 2.04, p = .15, $\eta^2$ = .007) nor in the justice-based pro-diversity beliefs condition (F(1, 299) = 0.30, p = .58, $\eta^2$ = .001).

Moreover, our results need to be further discussed in three regards. Besides a somewhat predictable main effect of instrumentality on the dependent variables (*F*(1, 298) = 8.25, *p* = .004, $\eta^2$ = .027; social distance: *F*(1, 301) = 7.34, *p* = .007, $\eta^2$ = .024), we found a negative main effect of justice-based pro-diversity beliefs (compared to instrumentality-based pro-diversity beliefs) on attitudes towards refugees prejudice (*F*(1, 298) = 5.11, *p* = .025, $\eta^2$ = .017; social dis-tance: *F*(1, 301) = 3.60, *p* = .059, $\eta^2$ = .012)–an effect we had not expected. In fact, we had expected that activating justice-based reasons to support refugees by, for example, referring to human rights, should lead to more *favorable* attitudes towards refugees–regardless of the actual instrumentality [26]. What we might have overlooked here though is that the political and societal climate in Germany during the time of data collection was characterized by a polariza-tion of political positions [27]. It is thus possible that emphasizing participants' moral obliga-tion to support refugees in the justice-based pro-diversity beliefs condition might have led participants to show reactance [28]. Second, despite some evidence for the idea that the effects of instrumentality-based pro-diversity beliefs are dependent on the actual instrumentality of diversity through the analyses of simple main effects, no significant interaction between type of pro-diversity beliefs and instrumentality emerged. One could speculate that being con-fronted with a positive or negative example of instrumentality (i.e., having a refugee home in a German village) did indeed influence participants' attitudes towards refugees (as indicated by significant main effects), but that this single example was not perceived as representative of ref-ugees' instrumentality in the whole society. For this reason, instrumentality may not have interacted with the activation of instrumentality-based pro-diversity beliefs (vs. justice-based

pro-diversity beliefs). Third, although none of the participants indicated in their open feedback that they guessed the purpose of the study one could criticize that demand characteristics in Study 2 could have driven the effects. In fact, an inflated relationship between the instrumentality manipulation and the dependent variables could have impeded an interaction effect of pro-diversity beliefs.

In Study 3, we conceptually replicated Study 2 but made some adjustments to the design: We refrained from using a single case example as a manipulation of general instrumentality of diversity and took additional measures to avoid demand characteristics. Moreover, although we used the same intergroup contexts as in Study 2, we utilized a less extreme manipulation of justice-based pro-diversity beliefs. Furthermore, Study 3 was conducted four years later in a climate that was less emotionally charged with regard to attitudes towards refugees.

## Study 3

Study 3 can be considered as a conceptual replication of Study 2. We, again, tested our predictions in the context of intergroup relations between Germans and refugees. Moreover, as in Study 2, we experimentally varied instrumentality (instrumental vs. detrimental) as well as type of pro-diversity beliefs (instrumentality-based vs. justice-based pro-diversity beliefs). However, we implemented five important changes compared to Study 2. First, we toned down the text manipulations for justice-based pro-diversity beliefs conditions–that is the references to justice-based foundations of helping refugees are less blatant than in Study 2. Second, data collection took place at a time in which the societal climate was less emotionally charged and politically polarized than during the timing of Study 1 in 2015, when Germany had just agreed to a large migration of refugees. Third, we made more efforts to reduce demand characteristics–that is the study was framed as a study on text comprehension. Participants received a total of three texts on current societal topics (i.e., global warming, house rental increases, and flight and migration) and were told that the texts were either presented as fully formulated texts or a list of bullet points. Participants assumed that the presentation order of the three texts and their format was randomly determined and the aim of the study was to compare participants' understanding of both forms of presentation. Fourth, the instrumentality manipulation was not restricted to a specific town but contained information about the usefulness of refugee immigration to Germany in general. Fifth, Study 3 was preregistered and adequately powered.

## Method

**Participants.**    Study 3 was preregistered (see https://osf.io/hvqcp). Data was collected online in December 2019 and January 2020. German participants were recruited via three crowdsourcing/online recruiting websites (i.e., Amazon's MTurk, Prolific, clickworker). Utilizing a conservative approach to determining sample size, we aimed for a sample size of 787 participants (based on small effect size of $f = .10$, $\alpha = .05$, and a power $(1-\beta)$ of .80). In total, 793 participants completed the questionnaire. We excluded 13 participants because they failed to correctly answer an attention check question. A minimum viewing time of 10 seconds was preprogrammed for the pages providing the texts that contained the manipulations. Hence, no participants were excluded due to not spending enough time on the respective pages.

Because we were interested in perceptions of ethnic majority members an additional 151 individuals were excluded because they had a migration background. The remaining 629 participants had a mean age of 32.09 years ($SD = 10.15$). The number of participants per condition ranged from 150 to 168.

**Procedure.** Study 3 was conducted online. It was announced as a survey on text comprehension. The study was conducted as a 2x2-design: As in Study 2, we manipulated two types of pro-diversity beliefs (instrumentality-based vs. justice-based pro-diversity beliefs) and the actual instrumentality of refugees for communities (instrumental vs. detrimental).

Participants first read an instruction text stating that the study dealt with text comprehension and that two types of texts would be compared, that is standard continuous text (as in newspaper articles) or text consisting of several bullet points (as in text presentation programs). Participants were told that the text version would be randomly determined. After that participants answered questions related to their age, nationality, occupation, and political orientation before they read the first text. This text served as a distractor text, dealt with issues on global warming, and was presented in a bullet point version for all participants. Next, participants answered questions regarding the comprehensibility, structure, and quality of this text as well as two questions about the context of the text. Similar items were also presented after participants read another distractor text that dealt with house rental increases that was written in a continuous text format. The third text constituted our manipulation, in that participants received a text about flight and migration. The text was written in a continuous format and contained information used to manipulate the two independent factors. In the first part of the text, participants either received information about positive consequences of previous mass-migration (instrumentality-based pro-diversity beliefs) or a reference to the Universal Declaration of Human Rights as well as the 1951 Refugee Convention (justice-based pro-diversity beliefs). Dependent on the conditions the respective paragraphs read:

Analyses of previous mass-migration movements reveal that nations can potentially benefit from the inclusion of refugees. Historically, state economic systems have often been shown to profit from migration and diversity.

Inclusion of refugees is an important part of the Universal Declaration of Human Rights. Moreover, the 1951 Refugee Convention provides legal support and social rights for refugees in the receiving countries. Germany and the European Union have signed both treaties.

The second part of the text, then, provided information on the actual instrumentality of refugees in Germany. It referred to a scientific report that either concluded that, effectively, refugee immigration in the last years can be regarded as beneficial (instrumental condition) or as a burden (detrimental condition) to the German society. The results of the report drew upon ostensible economic, organizational, and societal indicators such as tax revenues, use of vacant job and training positions, membership in communal clubs and associations to determine instrumental or detrimental effects, respectively. Subsequent to reading this text, participants were again presented with items on comprehensibility, structure, and quality of the text as well as two questions about the context of the text (of which one functioned as a manipulation check for the instrumentality manipulation). In a next step, we measured additional distractor items asking for participants' knowledge about the topic and whether the texts contained new information for them. After that, the dependent variable prejudice towards refugees was measured along with a number of items related to the contents of the other (distractor) texts. After having completed the questionnaire participants were thanked and debriefed.

**Measures.** *Prejudice towards refugees* was measured with the same four items used in Study 2 (e.g., 'There are too many refugees living in Germany', 'Refugees are a burden for the social welfare system'; $\alpha = .913$). The items were answered on a 7-point-scale ranging from 1 = *do not agree at all* to 7 = *totally agree*. The four items were aggregated to form a composite

**Table 6. Means and standard deviations of measures of Study 3.**

| | | | M | SD |
|---|---|---|---|---|
| prejudice tw. refugees | justice-based pro-diversity beliefs | instrumental | 3.05 | 1.77 |
| | | detrimental | 3.43 | 1.63 |
| | instrumental pro-diversity beliefs | instrumental | 3.04 | 1.73 |
| | | detrimental | 3.75 | 1.64 |
| | general | | 3.31 | 1.72 |
| political orientation | justice-based pro-diversity beliefs | instrumental | 3.21 | 1.23 |
| | | detrimental | 3.10 | 1.19 |
| | instrumental pro-diversity beliefs | instrumental | 3.16 | 1.25 |
| | | detrimental | 3.19 | 1.17 |
| | general | | 3.16 | 1.21 |
| manipulation check | instrumental conditions | 'Integration is a success story and Germany, in general, profits from refugees' | n = 305 | |
| | | 'Integration is anything else than a success story and Germany, in general, does not profit from refugees' | n = 21 | |
| | detrimental conditions | 'Integration is a success story and Germany, in general, profits from refugees' | n = 13 | |
| | | 'Integration is anything else than a success story and Germany, in general, does not profit from refugees' | n = 290 | |

score. A manipulation check was included comprising one dichotomous item ('What does the report reveal with regard to the integration of refugees?', response options: 'Integration is a success story and Germany, in general, profits from refugees' vs. 'Integration is anything but a success story and Germany, in general, does not profit from refugees').

Moreover, we measured *political orientation* on a 7-point-scale ranging from 1 = *left* to 7 = *right* as a potential covariate, alongside a number of unrelated items that served as distractor items.

## Results and discussion

Table 6 lists the descriptive statistics as well as intercorrelations between the measures. Note, that we additionally ran analyses prior to exclusion of participants with migration background. Results were comparable with the results for the reduced sample (S5 Table).

Correlations between political orientation and prejudice ($r = .602$, $p < .001$) were large. Hence, as in Study 2, we included political orientation as a covariate in the respective analyses [22] (see S6 Table).

A significant result of a $X^2$-test indicates that participants in the instrumental condition were more likely to choose the answer 'Integration is a success story and Germany, in general, profits from refugees' than 'Integration is anything else than a success story and Germany, in general, does not profit from refugees' and vice versa in the detrimental condition ($X^2 (1) = 500.63$, $p < .001$), confirming the effectiveness of our manipulation.

We then tested whether type of pro-diversity beliefs and instrumentality interacted: No significant interaction occurred ($F (1, 624) = 1.00$, $p = .318$, partial $\eta2 = .002$; full results are depicted in Table 7). Next, we ran simple main effects contrasting the factor instrumentality (instrumental vs. detrimental) with type of pro-diversity beliefs (i.e. instrumentality-based vs. justice-based). We observed a significant positive effect of the detrimental condition (compared to the instrumental condition) on prejudice in the instrumentality-based pro-diversity beliefs ($F (1, 624) = 21.27$, $p < .001$, partial $\eta2 = .033$) as well as in the justice-based pro-diversity beliefs conditions ($F (1, 624) = 9.49$, $p = .002$, partial $\eta2 = .015$).

**Table 7. Results of 2-factorial ANOVAs in Study 3.**

|  | prejudice | | | |
|---|---|---|---|---|
|  | *F* | *df* | *p* | partial $\eta^2$ |
| corrected model | 101.80 | 4 | .001 | .394 |
| constant | 16.90 | 1 | .001 | .026 |
| political orientation | 374.44 | 1 | .001 | .375 |
| pro-diversity beliefs (justice vs. instrumental) | 1.82 | 1 | .178 | .003 |
| instrumentality of refugees (instrumental vs. detrimental) | 29.409 | 1 | .001 | .045 |
| pro-diversity beliefs X instrumentality of refugees | 1.00 | 1 | .318 | .002 |
| error |  | 624 |  |  |
| $R^2$ |  |  |  | .394 |

Unexpectedly, and contrary to the results of Study 2, an effect of instrumentality on prejudice occurred independent of the type of pro-diversity beliefs that were activated. Note, however, that in line with our overall reasoning prejudice scores were marginally significantly different between both detrimental conditions ($F$ (1, 308) = 2.90, $p$ = .09, partial $\eta2$ = .009; controlled for covariate political orientation)–that is prejudice scores were marginally higher in the instrumentality-based pro-diversity and detrimental condition ($M$ = 3.75, $SD$ = 1.64) than in the justice-based pro-diversity and detrimental condition ($M$ = 3.43, $SD$ = 1.63). Fig 3 illustrates the results.

In sum, we were unable to find evidence for our assumption that positive effects of instrumentality-based pro-diversity beliefs (compared to justice-based pro-diversity beliefs) are dependent on the actual instrumentality of groups. However, a marginally significant difference between the type of pro-diversity beliefs emerged for outgroup attitudes when the presence of refugees was framed as detrimental. In other words, we found some weak evidence for a negative effect of negative instrumentality on outgroup attitudes when instrumentality-based pro-diversity beliefs (compared to justice-based pro-diversity beliefs) were made salient.

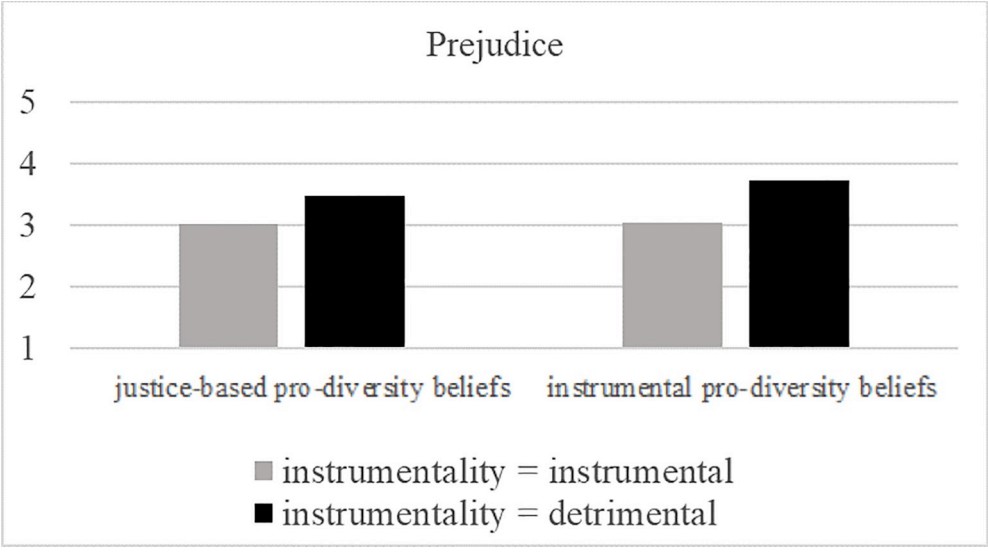

**Fig 3. Effects of pro-diversity beliefs (instrumentality- vs. justice-based pro-diversity beliefs) and instrumentality (instrumental vs. detrimental) on prejudice, and social distance in Study 3.** Covariate political orientation is evaluated at 3.16.

## Study 4

In Study 4, we tested our predictions in the context of intergroup relations between German students and international students at German universities. As in Studies 2 and 3, we manipulated type of pro-diversity beliefs and instrumentality. Intergroup attitudes were measured with items tapping prejudice towards exchange students.

## Method

**Participants.**   Study 4 was preregistered (see https://osf.io/cjpv3/). Please note, however, that two modifications of the preregistered design were made. The overall sample size of Study 4 differed from the planned sample size in the preregistration. This is due to the fact that we not only collected data at our university but also contracted a survey-company to gather data. Unfortunately, a large number of participants did not properly read the texts implemented in the questionnaire and were thus excluded (see preregistration). For this reason, we had to collect data from more participants than planned. Second, due to constraints regarding the questionnaire length we had to limit the number of items and discarded items measuring group salience. This decision was made after a non-editable version of the preregistration was created.

Data was collected between July and December 2016. In total, 785 current or former students completed the questionnaire. Of these, 23 participants failed to correctly answer at least one of two attention check questions and were excluded. In addition, 281 participants were excluded because they spent less than ten seconds on one or both of the pages that contained the manipulations (criteria based on duration of test runs; mean time spent on the page comprising the first manipulation was 50.29 seconds ($SD$ = 232.17), mean time spent on the page comprising the second manipulation was 62.27 seconds ($SD$ = 547.94). Because we studied attitudes towards immigrants that are exchange students, we considered it as reasonable to exclude participants that belonged to either one of these groups (i.e. participants that had a migration background or are/were exchange students themselves, see preregistration). Hence, 65 participants were excluded because they had a migration background. Moreover, we excluded an additional 71 participants who reported having studied abroad. Of the remaining 345 participants (mean age = 32.80, $SD$ = 9.71), 186 were women, 158 men, and one person indicated having another gender. Aiming for a sample including former and current students, we recruited participants from courses at our university and gathered data with the help of an external survey institute. Accordingly, 96 participants were psychology students who participated in return for course credit and 249 participants were recruited by the survey institute. The number of participants per condition ranged from 80 to 92.

**Procedure.**   Study 4 was conducted online and was announced as a survey on attitudes towards exchange study programs. Similar to Studies 2 and 3, it was conducted as a 2x2-design: We manipulated type of pro-diversity beliefs (instrumentality-based pro-diversity beliefs vs. justice-based pro-diversity beliefs), and the actual instrumentality of exchange students for German universities (instrumental vs. detrimental).

Participants first answered several demographic questions, items on contact experiences with foreign students and a single item asking for participants' political orientation. After that participants read two texts providing "background information on the topic of studying abroad". The texts contained manipulations of two independent factors. In the first text, participants received information on the legal background and the quantity of students studying abroad in the EU. Moreover, participants were presented a reason for why German universities should be motivated and take measures to receive foreign exchange students, activating

either instrumentality-based or justice-based pro-diversity beliefs (full manipulation texts are depicted in S3 File). Dependent on the conditions the respective paragraphs read:

> Prof. Wanka, the German Minister for Education and Research, claims that universities profit from internationalization–for example because the diversity of perspectives can improve teaching and research.

> Prof. Wanka, the German Minister for Education and Research, claims that that Germany allows foreign students to study at German universities because Germany is bound to support its EU partners: Only if all countries cooperate properly with regard to exchange study programs are we able to grant all EU citizens fair access to education.

Next, participants read another text dealing with the question of how internationalization affects the quality of teaching and research at German universities. This text was used to manipulate the actual instrumentality of exchange students. Participants were given fictitious statistics of a popular German university ranking. The text as well as a line chart either suggested a positive (instrumental condition) or a negative (detrimental condition) relationship between the number of exchange students at German universities and universities' quality of teaching and research, as well as student satisfaction. After having read the text participants answered a number of distractor items related to university policies, as well rating their own interest in and experiences with exchange study programs. Finally, participants answered items functioning as a manipulation check as well as items measuring the dependent variable, prejudice towards exchange students. Upon completing the questionnaire participants were thanked and debriefed.

**Measures.** All items were answered on 7-point-scales ranging from 1 = *do not agree at all* to 7 = *totally agree*. Items measuring the same construct were aggregated to form a composite score. *Prejudice towards exchange students* were measured with five self-generated items. However, we excluded one item ('Foreign exchange students are hard-working') because it decreased the scale's reliability. Hence, our prejudice measure comprised four items (e.g., 'I would not like to study at a university that hosts a lot of foreign exchange students', 'There are too many foreign students at German universities.'; $\alpha = .668$; $M = 2.74$, $SD = 0.99$; reliability prior to exclusion of item: $\alpha = .660$). Although exclusion of the item only slightly increased the reliability we considered the exclusion as important given the fact that the reliability falls below the critical threshold of $\alpha = .70$ [e.g., 29].

The manipulation check *perceived instrumentality* was measured with three items (e.g., 'German universities profit from exchange students.', 'The more international German universities are, the more satisfied the students.'; $\alpha = .839$; $M = 4.80$, $SD = 1.29$). Prejudice and perceived instrumentality were significantly correlated ($r = -.452$, $p < .001$). We measured *political orientation* prior to the manipulation, as a potential control variable. However, the correlation between political orientation and prejudice towards exchange students was moderate ($r = -.294$, $p < .001$). Hence, we refrained from including political orientation as a covariate [22]. Nevertheless, results with inclusion of this covariate can be found in the Supporting Material (S7 Table). They were comparable with the results obtained without the covariate political orientation.

Additionally, a number or unrelated items were included as distractor items (i.e. items focusing on university policies, interest in exchange study programs and contact with exchange students).

## Results and discussion

As mentioned above, data for this study included a student sample and a general population sample provided by an external survey institute. In a three-factorial ANOVA, we tested

**Table 8. Means and standard deviations of measures of Study 4.**

| | | | *M* | *SD* |
|---|---|---|---|---|
| perceived instrumentality | justice-based pro-diversity beliefs | instrumental | 5.15 | 1.20 |
| | | detrimental | 4.47 | 1.49 |
| | instrumental pro-diversity beliefs | instrumental | 5.30 | 0.90 |
| | | detrimental | 4.28 | 1.23 |
| | general | | 4.80 | 1.29 |
| prejudice | justice-based pro-diversity beliefs | instrumental | 2.72 | 1.03 |
| | | detrimental | 2.71 | 1.09 |
| | instrumental pro-diversity beliefs | instrumental | 2.57 | 0.81 |
| | | detrimental | 2.95 | 1.00 |
| | general | | 2.74 | 0.99 |

whether type of subsample directly affected our dependent variable prejudice or interacted with one or both of our manipulations. The type of subsample had a direct effect on prejudice ($F$ (1, 335) = 13.048, $p < .001$, partial $\eta^2$ = .037), showing higher prejudice scores in the general population sample ($M$ = 2.86, $SD$ = .96) compared to the university sample ($M$ = 2.42, $SD$ = 1.01). However, the type of subsample did not interact with the manipulations (pro-diversity beliefs: $F$ (1, 335) = 0.270, $p$ = .604, partial $\eta^2$ = .001; instrumentality: $F$ (1, 335) = 0.009, $p$ = .925, partial $\eta^2$ = .001; three-way interaction: $F$ (1, 335) = 1.339, $p$ = .248, partial $\eta^2$ = .004). Because no interaction occurred and the share of participants from subsamples was equally distributed across the experimental conditions ($p$'s < .084), we refrained from controlling for the influence of the subsample. The study was also administered along with another, unrelated study. The order of both studies was randomized. In an additional three-factorial ANOVA, we, hence, tested whether the order of studies affected our dependent variable prejudice or whether it interacted with one or both of our manipulations. No main or interaction effects occurred ($p$'s < .56). We therefore refrained from controlling for the order of studies.

Note that we additionally ran analyses prior to exclusion of participants with migration background and study experience abroad. Results somewhat differed from the results for the reduced sample (S8 Table).

Table 8 provides an overview of the descriptive statistics. We first tested whether our manipulation was successful. Perceived instrumentality should be influenced by the instrumentality manipulation but not by the type of pro-diversity beliefs manipulation. Results of a 2-factorial ANOVA supported these assumptions (main effect of type pro-diversity beliefs: $F$ (1, 341) = 0.035, $p$ = .85, partial $\eta^2$ = .000; main effect of instrumentality: $F$ (1, 341) = 41.822, $p < .001$, partial $\eta^2$ = .109; interaction effect: $F$ (1, 341) = 1.71, $p$ = .19, partial $\eta^2$ = .005). Perceived instrumentality was higher in the instrumental ($M$ = 5.23, $SD$ = 1.05) than in the detrimental ($M$ = 4.37, $SD$ = 1.46) conditions.

We found a marginally significant interaction between type of pro-diversity beliefs (i.e., instrumentality-based vs. justice-based pro-diversity beliefs) and instrumentality on prejudice ($F$(1, 339) = 3.49, $p$ = .06, $\eta^2$ = .010; full results are depicted in Table 9). We then tested simple main effects contrasting the factor instrumentality in dependence of pro-diversity beliefs (instrumentality-based vs. justice-based). In line with our assumptions, we observed a significant positive effect of the detrimental condition (compared to the instrumental condition) on prejudice in the instrumentality-based pro-diversity beliefs condition ($F$ (1, 339) = 6.94, $p$ = .009, partial $\eta^2$ = .020) but no effect in the justice-based pro-diversity condition ($F$ (1, 339) = 0.01, $p$ = .943, partial $\eta^2$ = .000). Note, however, that both detrimental conditions do not significantly differ from each other with regard to prejudice ($t$ (169) = -1.507, $p$ = .134)–although a

**Table 9. Results of 2-factorial ANOVAs in Study 4.**

| | prejudice | | | |
|---|---|---|---|---|
| | *F* | *df* | *p* | partial η² |
| corrected model | 2.36 | 3 | .072 | .020 |
| constant | 2647.61 | 1 | .001 | .886 |
| pro-diversity beliefs (justice-based vs. instrumental) | 0.16 | 1 | .693 | .000 |
| instrumentality of exchange students (instrumental vs. detrimental) | 3.11 | 1 | .079 | .009 |
| pro-diversity beliefs X instrumentality of exchange students | 3.49 | 1 | .063 | .010 |
| error | | 339 | | |
| *R²* | | | | .020 |

difference can be observed on a descriptive level (instrumentality-based pro-diversity beliefs detrimental condition: $M = 2.95$, $SD = 1.00$; justice-based pro-diversity beliefs detrimental condition: $M = 2.71$, $SD = 1.09$). Fig 4 gives an overview of the results.

As in Study 2, results of simple main effects analyses of Study 4 support the idea that effects of instrumentality-based pro-diversity beliefs are dependent on actual instrumentality. However, our assumption that presenting participants with instrumentality-based pro-diversity views surrounding exchange students increases negative attitudes towards this group when the presence of exchange students at universities was portrayed as having detrimental consequences can only be confirmed on a descriptive level. In other words, we did not find significant effects suggesting that activating instrumentality-based pro-diversity beliefs can lead to a deterioration of attitudes towards diverse outgroup members if diversity turns out to be detrimental rather than instrumental.

## General discussion

In recent debates one popular argument in favor of ethnic diversity is that groups and societies can profit from diversity because it facilitates creativity, improves problem-solving, and helps

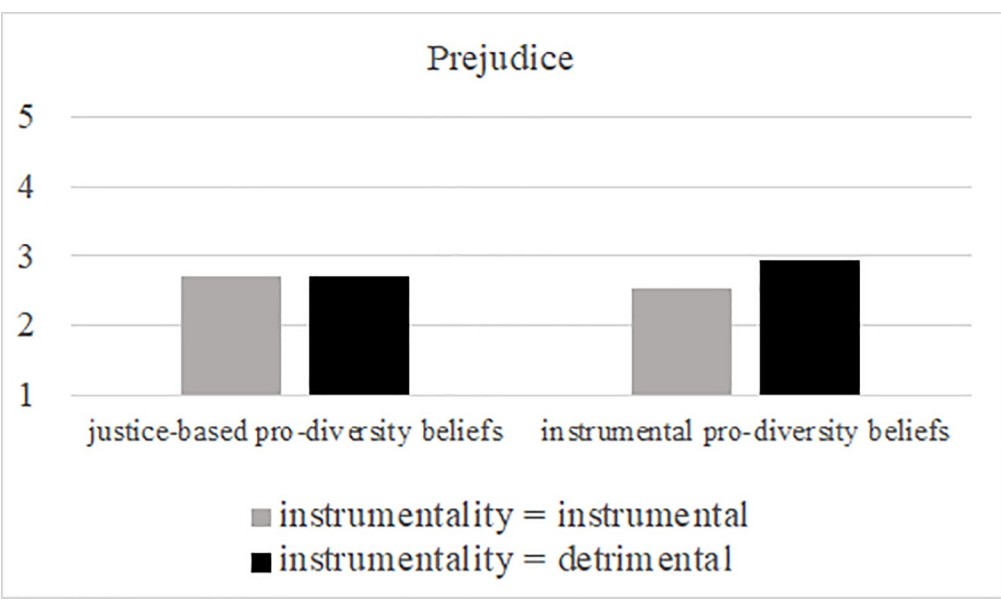

**Fig 4. Effects of pro-diversity beliefs (instrumentality- vs. justice-based pro-diversity beliefs) and instrumentality (instrumental vs. detrimental) on prejudice in Study 4.**

achieve new goals. This well-intentioned argument implies the normative view that diversity should be valued because it benefits the performance, success and functioning of groups. Our research critically evaluated the consequences of holding such instrumentality-based pro-diversity beliefs. We specifically tested whether the actual instrumentality of diverse groups influences the relationship between instrumentality-based pro-diversity beliefs and intergroup attitudes.

Across a set of four experimental studies with varying operationalizations of intergroup attitudes, we did not find systematic evidence that the perceived non-instrumentality of diverse groups (i.e. perceptions of detrimental effects of diversity)–compared to instrumentality (i.e. instrumental effects of diversity)–reduced or even reversed the positive effects of instrumentality-based pro-diversity. At best, one could argue that we found sporadic and weak support for our assumptions. In a context involving direct collaboration within diverse groups (Study 1), we only found weak evidence for a mitigating effect of participants' collaboration in diverse groups that resulted in detrimental group performance on the positive relationship between instrumentality-based pro-diversity beliefs and favorable outgroup attitudes. Importantly, the robustness of this finding is challenged by the fact that we found no significant interaction effect when comparing the influence of detrimental interactions within diverse vs. non-diverse control groups for two of the three dependent variables. Based on this finding, we cannot rule out a mere effect of negative feedback–independent of the diversity of the group in which the interaction took place. Likewise, results of Studies 2 and 4 only provided weak evidence for the hypothesis that a positive effect of instrumentality-based pro-diversity beliefs is dependent on the salience of positive instrumentality. However, in both studies, no negative effect of instrumentality-based pro-diversity beliefs (compared to non-instrumentality-based pro-diversity beliefs) after detrimental outcomes could be observed. In Study 3, the idea that positive effects of instrumentality-based pro-diversity beliefs are dependent on the actual instrumentality of groups was not supported. We did however obtain some (marginally significant) evidence that negative instrumentality has a stronger effect on negative outgroup attitudes when instrumentality-based pro-diversity beliefs (vs. non-instrumentality-based pro-diversity beliefs) were made salient.

Summarizing our results, we did not find conclusive evidence for our ideas. We cannot say with certainty whether the lack of support for our hypotheses is a consequence of methodological shortcomings or whether evidence for the underlying theoretical assumptions are false. One could argue, however, that we found some preliminary evidence to suggest that when diversity is perceived as detrimental, holding or being presented with instrumentality-based pro-diversity beliefs might lead to negative effects on outgroup attitudes or weaken positive effects on outgroup attitudes. Some of our findings suggest that the actual or perceived instrumentality of diversity might matter, i.e. that instrumentality-based pro-diversity beliefs do not unequivocally bring about desired effects when diversity is not instrumental. Because of our mixed findings, it is, however, evident that more research needs to be done to further support the sporadic evidence we found throughout our studies.

Future research should also try to answer the question *why* detrimental diversity might mitigate positive effects of instrumentality-based pro-diversity beliefs. One obvious explanation is that an instrumentality-based rationale induces, increases, or helps to legitimize (existing) rejection of outgroup members [6]. One could also argue that beliefs in diversity shape individuals' perceptions of diverse groups in such a way that subgroups become particularly salient [8, 12]–especially if outgroup members do not fulfill ingroup members' expectations [30]. In line with this reasoning, Wolsko, Park, Judd, and Wittenbrink [31] showed that acknowledging diversity leads to increased salience of groups in intergroup encounters. Moreover, Vorauer and Sasaki [32] showed that focusing on differences between groups can lead to more negative

intergroup behavior when interactions are perceived as negative. In a similar vein, Paolini, Harwood, and Rubin [33] demonstrated that negative intergroup contact increases the salience of interaction partners' group memberships and, hence, leads to negative outgroup attitudes [34]. Linking these findings with our work on pro-diversity beliefs, it is thus possible that for people attuned to instrumentality-based views on diversity, detrimental cooperation or interaction in diverse groups are perceived as negative intergroup contact experiences and therefore lead to higher subgroup salience and less positive outgroup attitudes. It remains, however, for future research to empirically confirm this idea. On a related note, one could criticize that the present studies are unable to disentangle processes related to the instrumentality of diverse groups from more basic processes related to experiences with ethnic outgroup members. In other words, an alternative explanation for our findings could be that instrumentality-based pro-diversity beliefs interact with retroactive evaluations of interactions as positive and negative intergroup contact [33]. Accordingly, more research is needed that addresses the interplay between instrumentality and more general valence of intergroup contact.

Although, our results did not provide robust evidence for out assumptions we consider the theoretical criticisms on the so-called business case for diversity as legitimate and relevant for practitioners. Focusing on organizational diversity policies, Dickens [35] warned that adapting a good-for-business logic can, under certain circumstances, lead to a business case *against* diversity. Support of diversity that is based on rational considerations about the instrumentality of diversity holds the risk of hampering or even harming intergroup relations if diversity ends up not being beneficial to group performance and success. Of course, we do not want to negate the importance of diversity for driving the success of groups per se, but our findings allow us to caution against over-emphasizing the instrumentality of diversity when encouraging pro-diversity views as a diversity-management strategy. Practitioners should therefore consider placing less emphasis, or at least not the sole focus on instrumentality-focused arguments in favor of diversity. Moreover, political supporters of ethnic diversity or immigration should keep in mind that well-intentioned arguments underlining the instrumental value in diversity can have contradictory effects. Indeed, recent surges in right-wing populism in the US and Europe are, in part, accompanied by instrumentality-focused ideas that immigrants should be opposed if they cannot contribute to the (economic) wealth of societies [21].

If future research brings about additional support for negative effects of instrumentality-based pro-diversity beliefs it would also be interesting to additionally study which message beyond instrumentality-based views should underlie the promotion of diversity. Reicher, Cassidy, Wolpert, Hopkins, and Levine [36], for example, argue that, besides instrumental considerations, social solidarity can also be driven by normative and identity-related views. On the one hand, they thus claim that solidarity arises when it is perceived as a normative part of the ingroup's social identity. Critics on the business case for diversity would probably agree with this idea and advise building a case for diversity on the grounds of equality-based and moral arguments [1]. However, in Study 2 we found a prejudice-increasing effect of morality-based pro-diversity beliefs (compared to instrumentality-based pro-diversity beliefs). We did not however find the same effect in Studies 3 and 4, although morality- and justice-based pro-diversity beliefs did also not reduce prejudice in these studies. Accordingly, it is questionable to assume that moral arguments in favor of diversity can be considered as a panacea when it comes to support for diversity and ethnic outgroups. At least in times of heated political debates, strategies involving the promotion of diversity on moral grounds can backfire and may even lead individuals to more strongly oppose diversity [27]. A potentially more promising alternative could involve the promotion of more inclusive social identity processes. Reicher and colleagues [35] propose that social solidarity can be triggered by seeing outgroups as members of a superordinate group, based on ideas underlying the common ingroup identity model

[37]. In line with this idea, Waldzus, Mummendey, Wenzel, and Weber [38] showed that devaluation of outgroups is reduced when in- and outgroups contribute to a complex proto-type of a joint superordinate group. Relatedly, Guerra and colleagues [16] showed that immigrant groups are seen as more positive if they not only contribute to the functioning of society but if they are also seen as an indispensable part of a superordinate identity. However, more research on the optimal strategies for promotion of pro-diversity beliefs, the content thereof, and their consequences is needed.

**Limitations and future directions.** Our study is the first to examine the interplay between instrumentality-based pro-diversity beliefs and actual instrumentality of diversity in driving intergroup relations, to empirically test criticisms on the business case for diversity. Despite the weak evidence for our assumptions, the studies in this paper nonetheless provide an important first step towards a deeper understanding of positive and negative consequences of frequently used instrumentality-based arguments in favor of organizational and societal diversity. However, we acknowledge a number of limitations that should be addressed in future studies. First, with regard to generalizability one should keep in mind that studies are based on data from highly-educated non-immigrant Germans, and all studies were situated in the context of ethnic diversity in Germany. Moreover, all studies were conducted online. We suggest that future research seeks to replicate our studies with larger (and more heterogeneous) samples, in different countries, and with both field- and lab-experimental methods.

On a different note, although we pre-registered Studies 3 and 4, we regret not having been able to do the same for Studies 1 and 2 as the awareness of the importance of Open Science measures emerged only in the later stages of the present research project. As an additional measure aimed at improving scientific standards, however, all data files are available at https://osf.io/kyxvf. Questionnaires can be found in the Supporting Material (see S3 File).

Moreover, we consider it important for future research to test our assumptions in real life collaborations in diverse groups. We recommend that future studies draw on existing diverse groups and study the interplay between beliefs in the instrumentality of diversity of studied groups and measures of (perceived) productivity/performance of these groups in shaping group relations and conflict.

Two additional suggestions for future research have already been touched upon above. First, future studies should address different pro-diversity measures and their content beyond purely instrumental underpinnings, and study their effectiveness in reducing prejudice and fostering cooperation and positive group relations. Based on our results, we thus consider it important to address the ideological and moral foundation of these approaches (e.g., instrumentality-based arguments, or an equality founded rationale, or an alternative to both paradigms; [1]). Second, future research should aim to get a better idea of the *processes* underlying some of our findings, to study potential mediators. Moreover, long term studies could help to shed light on potential bi-directional causal processes between (non-) instrumentality and instrumentality-based pro-diversity beliefs over time.

## Supporting information

**S1 File. Pretest results.**
(PDF)

**S2 File. Results for additional outgroups.**
(PDF)

**S3 File. Questionnaire items and texts for all studies.**
(PDF)

**S1 Table. Results of Study 1 with inclusion of covariate political orientation.**
(PDF)

**S2 Table. Results of Study 1 without exclusion of participants with migration background.**
(PDF)

**S3 Table. Results of Study 2 without exclusion of participants with migration background.**
(PDF)

**S4 Table. Results of Study 2 without inclusion of covariate political orientation.**
(PDF)

**S5 Table. Results of Study 3 without exclusion of participants with migration background.**
(PDF)

**S6 Table. Results of Study 3 without inclusion of covariate political orientation.**
(PDF)

**S7 Table. Results of Study 4 with inclusion of covariate political orientation.**
(PDF)

**S8 Table. Results of Study 4 without exclusion of participants with migration background and participants that studied abroad.**
(PDF)

## Author Contributions

**Conceptualization:** Mathias Kauff.

**Methodology:** Oliver Christ.

**Project administration:** Mathias Kauff.

**Writing – original draft:** Mathias Kauff.

**Writing – review & editing:** Mathias Kauff, Katharina Schmid, Oliver Christ.

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
