## [Decision Letter · Decision Letter 0]

31 Oct 2019

PONE-D-19-24506

When good for business is not good enough: Effects of pro-diversity beliefs and instrumentality of diversity on intergroup attitudes

PLOS ONE

Dear Dr. Kauff,

I have now received two reviews of your manuscript and also read it myself. Although the reviewers and I believe that you are addressing an interesting topic, we have several concerns that prevent us from recommending this paper for publication. As you acknowledge in the general discussion, the crucial effects were quite weak. For me, this problem is aggravated by a number of issues. First, only Experiment 3 was preregistered. Because there is a lot of data trimming in all of the experiments, questions can be raised about researcher degrees of freedom. I appreciate that you report the outcome of analyses with different subsets of participants, but sometimes these analyses also reveal different results, which raises doubts about the robustness of the effects. Second, in Experiments 1 and 2, you test single effects and draw conclusions on the basis of the fact that one effect is significant whereas another is not. In my opinion, it is more appropriate to compare effects directly. For instance, in Experiment 1, rather than emphasizing that “detrimental diversity (compared to instrumental diversity) had a mitigating effect on the positive relation between pro-diversity beliefs and general outgroup attitudes” whereas “neither non-instrumentality in diverse (neutral diverse vs. instrumental diverse condition; b = -0.610, SE = 0.487, p = .213, CI95% = -1.576, .355) nor negative instrumentality in non-diverse groups (detrimental non-diverse vs. instrumental diverse condition; b = -0.800, SE = 0.607, p = .191, CI95% = -2.005, 0.405) interacted with pro-diversity beliefs” it would be better to test whether the effect for detrimental diversity was significantly different than that of non-instrumentality in diverse or negative instrumentality in non-diverse groups. Also, to control for effects of negativity, a direct comparison of detrimental diversity and detrimental non-diverse conditions is needed. This concern relates to Comment 8 of Reviewer 1, who notes that in Experiment 3, the comparison with detrimental consequences under justice-based pro-diversity beliefs is crucial but not reported. A quick inspection of the data suggests that these crucial comparisons will generate effects that are even weaker than the effects that you report now.

For these and other reasons, I doubt whether the conclusions that you draw, are actually warranted by your data. Although PLoS ONE is not concerned primarily with the level of contribution, we do want to make sure that the papers we publish report data that are likely to be replicable and conclusions that do not go beyond the data. I’m afraid that your paper currently does not meet these criteria. In order to reach the criteria, not only additional analyses are needed but also new data from a new preregistered study. Although I cannot accept the current version of your paper for publication, I would be willing to consider a new paper that reports a new study as well as a more appropriate analyses of all data (old studies and new study), and that reports conclusions that are warranted by the data, even if the conclusion is that there is no support for your hypotheses. The question you address is an interesting one that has implications for the real world, so it is important that we get this right.

Here are a number of other thoughts that I had while reading your paper:

Ethics statement: “Studies were approved by the local ethics committee of the Psychology Department at the University of Marburg (Germany, 2013-24k).”: If you decide to resubmit, please include a copy of your application for ethical approval, as well as a the letter in which the Ethics Committee states their approval and the conditions for their approval. It would be great if you could also make these documents available on the OSF. I think this is important because your studies do have aspects that are potentially problematic from an research ethics point of view (e.g., collecting sensitive personal data such as political preference, providing potentially controversial information such as opinions of a minister and so-called information about the success of integration of immigrants and the impact of visiting students on the quality of education). Even if there are reasons for not sharing these documents, it would be important to provide in the manuscript more information about the debriefing that you used, as well as the way you dealt with personal information during the analyses and data storage.Data availability: You state that “All 4 files are available from the OSF database (https://osf.io/kyxvf/)”. However, they are currently available only after asking for permission. Please note that the reviewers and I need to have access to the data now, that is,  before the paper can be accepted for publication. Also, unless there are reasons for not making the data available to all without having to ask for permission, please confirm that all restrictions will be removed after acceptance.P. 15: “ In line with our predictions, the effect of pro-diversity beliefs was smallest in the detrimental diverse condition”: This is true in absolute terms but you did not perform the statistical tests necessary to back up this claim. Other example, p. 16: “the magnitude of the positive effect of pro-diversity beliefs on all dependent variables was smaller when participants’ collaboration in diverse groups resulted in detrimental group performance”. In sum, please refrain from interpreting differences in absolute scores.P. 20: “Correlations between political orientation and prejudice (r = .533, p < .001) as well as social distance (r = .464, p < .001) were large. Hence, we included political orientation as a covariate in the respective analyses”: Is the mere fact of having high correlations sufficient to include political orientation as a covariate? I guess there must be some theoretical reason for doing this? Providing a strong justification for this is particularly important given that it changed the results.P. 24: “Moreover, we excluded an additional 71 participants who reported having studied abroad”: Was this pre-registered? If so, why did you decide to also run an analysis that included the data of these participants (see p. 27)?P. 28: “moderation of type of pro-diversity beliefs (i.e., instrumentality-based vs. justice-based pro-diversity beliefs) on prejudice by instrumentality”: difficult to understand. If possible, please clarify.

If you decide to resubmit a revised version of your paper, please also take into account all the comments of the reviewers.

We would appreciate receiving your revised manuscript by Dec 15 2019 11:59PM. To enhance the reproducibility of your results, we recommend that if applicable you deposit your laboratory protocols in protocols.io, where a protocol can be assigned its own identifier (DOI) such that it can be cited independently in the future. For instructions see: http://journals.plos.org/plosone/s/submission-guidelines#loc-laboratory-protocols

We look forward to receiving your revised manuscript.

Kind regards,

Jan De Houwer

Academic Editor

PLOS ONE

Journal Requirements:

Reviewers' comments:

Reviewer's Responses to Questions

**Comments to the Author**

1. Is the manuscript technically sound, and do the data support the conclusions?

Reviewer #1: Yes

Reviewer #2: Partly

2. Has the statistical analysis been performed appropriately and rigorously? 

Reviewer #1: Yes

Reviewer #2: Yes

3. Have the authors made all data underlying the findings in their manuscript fully available?

Reviewer #1: Yes

Reviewer #2: Yes

4. Is the manuscript presented in an intelligible fashion and written in standard English?

Reviewer #1: Yes

Reviewer #2: Yes

5. Review Comments to the Author

Reviewer #1: The paper challenges conventional wisdom in the area of diversity research by suggesting that instrumentality-based pro-diversity beliefs can backfire when diversity turns out to be detrimental. Three studies provide some evidence that when diversity yields detrimental consequences, instrumentality-based pro-diversity beliefs can weaken positive attitudes towards ethnic outgroups, or even create more negative attitudes compared to situations in which there are morality-based pro-diversity beliefs. The paper is well written, the studies are well conducted, and the findings are thought provoking. In the following I offer a number of suggestions to further improve the paper in chronological order, but comments 6 and 8 (and potentially 9) probably demand most attention.

1. Overall framing: I think your point would be even clearer when you refer to it as a backlash effect from instrumentality-based pro-diversity beliefs, and would also explicitly call it like that in the title. That way your main point is clearer.

2. I like and understand the positioning of the paper in the diversity beliefs literature, but one of the main questions I have is whether your findings are really due to experiences with diversity, or due to experiences with ethnic outgroup members. This is difficult/impossible to disentangle in your studies, but it does require discussing the literature on the contact hypothesis (e.g., Allport, 1954; Tropp & Pettigrew, 2005) and either providing an integrated perspective and argument, or discussing it as an alternative explanation for your findings. I guess the latter option would be easier, but the first could be more exciting and may actually not be too difficult given that the general idea of such an integration would be that positive [negative] experiences with a group of diverse members would positively [negatively] shape attitudes towards those members.

3. Your intro and theory sections are well-written, but the comparison with pro-morality beliefs came too late for me. In the organizational literature, instrumentality-based pro-diversity beliefs are usually contrasted with instrumentality-based pro-similarity beliefs, so I was also surprised that you used a different comparison category. So it would be helpful if you introduce and explain morality-based pro-diversity beliefs earlier on as the comparison category.

4. On p. 13, you discuss your operationalization of pro-diversity beliefs (IV) and of attitudes towards immigrants (DV ) in one paragraph. Please put them in different paragraphs to make it easier for the reader to understand.

5. In study 2, you excluded participants when they spent less than 10 seconds on both pages explaining the context, but the average time spent on those pages was much higher. So isn't 10 seconds still way too short and shouldn't the inclusion criteria be stricter (e.g., spending at least 25 seconds in order to have read it properly)?

6. To me, the design of study 2 came across as very obvious given that you measured participants' prejudice directly after the manipulation. It is also weird that the distractor items only came after measuring the DV, which means that there was little use of the distractor items for the study. So did you correctly write down the order of the study here, and if so, can you explain why you did it in this order and how you can be sure that participants were not aware of what your aims were in this study? And if it was too obvious, perhaps that also explains why you didn't find an interaction effect, given that the link between the manipulation and the DV was too strong? Either way, I think it would be good to critically consider whether the second study is of sufficient quality.

7. In study 3, the reliability of your prejudice measure is below the standard threshold of a = .70. That does not necessarily constitute a problem, but at the very least you should point it out. Does the removal of an item yield a better score?

8. In terms of interpretation, on p. 28 at the bottom you suggest that "presenting participants with instrumentality-based pro-diversity views surrounding exchange students increases negative attitudes towards this group when the presence of exchange students at universities was portrayed as having detrimental consequences". I understand that interpretation based on the comparison with the situation when the consequences were instrumental, but I wonder if that's the correct comparison. Shouldn't the comparison be with detrimental consequences under justice-based pro-diversity beliefs? In case instrumental pro-diversity beliefs even under detrimental consequences of diversity do not yield more negative attitudes towards outgroup members, then overall there is no problem with instrumental pro-diversity beliefs. However, in case instrumental pro-diversity beliefs do yield more negative attitudes under detrimental consequences compared to morality- or justice-based pro-diversity beliefs, then it really becomes questionable if instrumental pro-diversity beliefs are desirable. In other places (e.g., p. 30, the paragraph starting with "summarizing") your interpretations of your findings go into a similar direction, where I thus wonder if they shouldn't focus more on whether or not instrumentality-based pro-diversity beliefs yield more negative attitudes compared to morality-based pro-diversity beliefs.

9. On a related point, you already mention in study 2 that you were somewhat surprised by the overall negative attitudes of those in a morality-based pro-diversity beliefs condition, and in general your findings do seem to indicate that when diversity is instrumental, instrumentality-based pro-diversity beliefs yield much more positive attitudes compared to morality-based pro-diversity beliefs. So you could also consider whether the actual story of the paper isn't that instrumental pro-diversity beliefs yield more positive/less negative attitudes when diversity also turns out to be instrumental - especially if it doesn't matter much whether people hold instrumentality- or morality-based pro-diversity beliefs when diversity is detrimental. Interestingly, such a conclusion would yield radically different implications. So I am looking forward to your reply.

Again, I really like the overall aim of the paper and hope these comments help to further bolster it's main message - whatever that in the end may be.

References

Allport, G. W. (1954). The nature of prejudice. New York: Addison-Wesley.

Tropp, L. R., & Pettigrew, T. F. (2005). Relationships between intergroup contact and prejudice among minority and majority status groups. Psychological Science, 16(12), 951-957.

Reviewer #2: The manuscript ‘When good for business is not good enough: Effects of pro-diversity beliefs and instrumentality of diversity on intergroup attitudes’ explores whether the positive effect of pro-diversity beliefs on general attitudes (as well as ratings of warmth and competence) of ethnic outgroups is moderated by the actual instrumentality of diversity. Specifically, the authors postulate that non-instrumental diversity weakens or even reverses the otherwise positive effect of pro-diversity believes on attitudes towards ethnic outgroups. A question relevant to research and practitioners alike.

I believe that the manuscript is a timely contribution addressing an issue of high practical relevance. I particularly appreciate that the authors pre-registered Study 3, that they report deviations from the pre-registration, and that they acknowledge some of the degrees of freedom typically faced by researchers. E.g., they report that the results are comparable including or excluding political orientation as covariate (S1, S3) and they clearly state differences in results emerging by in- or excluding the covariate (in S2).

While I have not recalculated the power analysis, I am little bit surprised that a sample of 100 participants (approx. 25 per cell) in Study 1 is sufficient for a moderation analysis. In light of the rather small sample, the authors might emphasize the marginal significant interactions too much (S1 warmth and competence), which are discussed as theoretical informative not only in Study 1 but also in Study 3 (even though I appreciated that the authors discuss this in the limitation section).

I further wonder why the authors choose instrumental-diverse and not the neutral condition as baseline.

Additional concerns

- Study 1: The authors state that the items were assessed on a 5-point-Likert scale, yet the mean for general attitudes is reported as 6.31 (Table 2). Please correct this inconsistency. Please further discuss the overall very positive attitudes toward immigrants

- Study 2: Please report the time frame of data collection

- Please discuss the unexpected finding that morality-based diversity beliefs lead to detrimental effects in greater details.

- Please report by how much the two deleted items (S1 and S3) affected the reliability of the scale

6. PLOS authors have the option to publish the peer review history of their article (what does this mean?). If published, this will include your full peer review and any attached files.

Reviewer #1: Yes: Hans van Dijk

Reviewer #2: No

---

## [Author Response · Author response to Decision Letter 0]

6 Mar 2020

Dear Professor De Houwer, 

Thank you very much for your constructive and encouraging comments and those of the two reviewers. We appreciate the suggestions, and feel that they have helped us to improve the clarity of the paper. We have carefully revised the paper and detail our response to each of the points raised below. We begin by responding to your editorial comments and then move to those of the two reviewers.

Please also note that we revised the supporting information – that is we have condensed the text and, now, present more tables. We hope that this leads to a greater clarity of the supporting information. 

Thank you again for your valuable suggestions and for giving us the chance to revise our manuscript. Below you will find the detailed description of the changes we made to the paper, and should you have any additional questions we will be more than happy to answer these.

Sincerely,

[the authors]

Editorial comments:

1. In Experiments 1 and 2, you test single effects and draw conclusions on the basis of the fact that one effect is significant whereas another is not. In my opinion, it is more appropriate to compare effects directly. For instance, in Experiment 1, rather than emphasizing that “detrimental diversity (compared to instrumental diversity) had a mitigating effect on the positive relation between pro-diversity beliefs and general outgroup attitudes” whereas “neither non-instrumentality in diverse (neutral diverse vs. instrumental diverse condition; b = -0.610, SE = 0.487, p = .213, CI95% = -1.576, .355) nor negative instrumentality in non-diverse groups (detrimental non-diverse vs. instrumental diverse condition; b = -0.800, SE = 0.607, p = .191, CI95% = -2.005, 0.405) interacted with pro-diversity beliefs” it would be better to test whether the effect for detrimental diversity was significantly different than that of non-instrumentality in diverse or negative instrumentality in non-diverse groups. Also, to control for effects of negativity, a direct comparison of detrimental diversity and detrimental non-diverse conditions is needed. 

Thank you very much for this important critique. When reviewing the Results section of Study 1, we noticed that our write-up thereof lacked some clarity and that we should have specified our methodological approach in more detail, which we have now done. In fact, we believe, that our approach is mostly in line with your ideas and recommendation, in that we do directly compare effects. 

We have now made this clearer by explaining in greater detail how we test our assumptions and justify why we chose to test detrimental diversity as well as neutral diversity against instrumental diversity. The respective paragraph reads: 

“We tested whether the effect of instrumentality-based pro-diversity beliefs on attitudes towards immigrants (i.e. general attitudes, warmth and competence) was moderated by instrumentality in diverse groups (dummy coded with three variables: a) instrumental diverse as a baseline condition vs. neutral diverse, b) instrumental diverse vs. detrimental diverse, and c) instrumental diverse vs. detrimental non-diverse). Dummy-coding in regression analyses with a multicategorical moderator allows for testing the moderation effect of each category (here: experimental condition) against a preassigned reference group [e.g., XYZ]. We assume that previously found prejudice-reducing effects of instrumentality-based pro-diversity beliefs occurred because diversity was perceived as instrumental. Building on this assumption, we used the instrumental diverse condition as a reference group to contrast the effects on non-instrumentality (neutral diverse condition) and negative instrumentality (detrimental diverse and detrimental non-diverse conditions) with positive instrumentality (instrumental diverse).” (p. 17).

Moreover, we provide more explanation regarding the outcomes of the analyses: 

“In this section, we focus on the interaction effects. Full results are, however, depicted in Table 3. Testing the detrimental diverse condition against the instrumental diverse condition, we observed a significant interaction effect of negative instrumentality on the relationship between instrumentality-based pro-diversity beliefs and general outgroup attitudes (b = 0.956, SE = 0.405, p = .020, CI95% = -1.759, -0.153). In other words, detrimental diversity (compared to instrumental diversity) had a mitigating effect on the positive relation between instrumentality-based pro-diversity beliefs and general outgroup attitudes. Neither non-instrumentality in diverse (i.e., neutral diverse vs. instrumental diverse condition; b = 0.610, SE = 0.487, p = .213, CI95% = -1.576, .355) nor negative instrumentality in non-diverse groups (i.e., detrimental non-diverse vs. instrumental diverse condition; b = 0.800, SE = 0.607, p = .191, CI95% = -2.005, 0.405) interacted with instrumentality-based pro-diversity beliefs. In other words, neither a neutral outcome (non-instrumental but not detrimental) in a diverse group nor a negative detrimental outcome in a non-diverse condition had an effect on instrumentality-based pro-diversity beliefs and general outgroup attitudes.” (pp. 17-18)

What was lacking in the previous version of the manuscript was a direct comparison between the detrimental diverse condition and detrimental non-diverse condition. Thank you very much for pointing this out. We now added the respective analyses as well as a discussion of the results of these analyses to the Results section of Study 1. 

“To test whether the interaction effect of non-instrumentality on the relationship between instrumentality-based pro-diversity beliefs and general outgroup attitudes was specific to interactions within a diverse group we additionally contrasted the detrimental diverse condition with the detrimental non-diverse condition as a moderator in a moderated regression. A contrast of both conditions did not significantly moderate the effect of instrumentality-based pro-diversity beliefs on general outgroup attitudes (b = -.156, SE = 0.666, p = .816, CI95% = -1.492, 1.180) – the effect of instrumentality-based pro-diversity beliefs on general outgroup attitudes did not differ in dependence of whether a detrimental interaction occurred in a diverse or a homogenous group” (p. 18-19)

2. This concern relates to Comment 8 of Reviewer 1, who notes that in Experiment 3, the comparison with detrimental consequences under justice-based pro-diversity beliefs is crucial but not reported. A quick inspection of the data suggests that these crucial comparisons will generate effects that are even weaker than the effects that you report now.

Thank you and Reviewer 1 for bringing up this important point. In the original version of the manuscript, we did not consider this comparison as a crucial comparison. However, because we not only argue that the prejudice-reducing effect of instrumentality-based pro-diversity is dependent on the actual instrumentality of diversity but also imply that a potential prejudice-reducing effect of instrumentality-based pro-diversity might occur under the impression of detrimental interaction within diverse groups we agree that it is important to present the outcome of this comparison. 

We now report and discuss the respective analyses for Studies 2 to 4.

3. I doubt whether the conclusions that you draw, are actually warranted by your data. Although PLoS ONE is not concerned primarily with the level of contribution, we do want to make sure that the papers we publish report data that are likely to be replicable and conclusions that do not go beyond the data. I’m afraid that your paper currently does not meet these criteria. In order to reach the criteria, not only additional analyses are needed but also new data from a new preregistered study. Although I cannot accept the current version of your paper for publication, I would be willing to consider a new paper that reports a new study as well as a more appropriate analyses of all data (old studies and new study), and that reports conclusions that are warranted by the data, even if the conclusion is that there is no support for your hypotheses. The question you address is an interesting one that has implications for the real world, so it is important that we get this right 

Thank you for your positive evaluation of the importance of our research question, for your critical take on our results and their interpretation, and, finally, for giving us the chance to conduct and include an additional study. We feel that this comment in particular helped us to improve the clarity and impact of our paper. 

First of all, as you will notice when re-reading the manuscript, we toned down the manuscript’s language with regard to interpretation and discussion of findings as well as their conclusions. 

More importantly, however, we now report an additional pre-registered study with a larger sample and a less obvious design (see Reviewer 1’s comment #6). This new study can be considered as a conceptual replication of Study 2 (and is, hence, presented as Study 3 in the revised version of the manuscript). Importantly, however, in response to your helpful suggestions, the study addresses a number of shortcomings of Study 2 – that is the rather narrow manipulation of instrumentality, potential demand characteristics, a rather extreme manipulation of morality-based pro-diversity beliefs, and data collection during a time where a prevalent political zeitgeist influenced by the so-called refugee crisis in Germany could have heavily influenced the results (see pp. 33-34)

4. Ethics statement: “Studies were approved by the local ethics committee of the Psychology Department at the University of Marburg (Germany, 2013-24k).”: If you decide to resubmit, please include a copy of your application for ethical approval, as well as a the letter in which the Ethics Committee states their approval and the conditions for their approval. It would be great if you could also make these documents available on the OSF. I think this is important because your studies do have aspects that are potentially problematic from an research ethics point of view (e.g., collecting sensitive personal data such as political preference, providing potentially controversial information such as opinions of a minister and so-called information about the success of integration of immigrants and the impact of visiting students on the quality of education). Even if there are reasons for not sharing these documents, it would be important to provide in the manuscript more information about the debriefing that you used, as well as the way you dealt with personal information during the analyses and data storage.

Data availability: You state that “All 4 files are available from the OSF database (https://osf.io/kyxvf/)”. However, they are currently available only after asking for permission. Please note that the reviewers and I need to have access to the data now, that is, before the paper can be accepted for publication. Also, unless there are reasons for not making the data available to all without having to ask for permission, please confirm that all restrictions will be removed after acceptance.

Thank you for these remarks. We now uploaded the ethics application as well as the ethics statement on OSF. Both documents are in German, however. We, hence, added the following information to the Method section of Study 1: 

“In all studies, written informed consent had to be given online on the first page, that is participants had to actively agree to take part in the study after having received information about the study, data handling, and risks of participation. Participants were aware that they could withdraw at any time without consequences. Data were completely anonymized before data storage. Written debriefings were given on the final page. We used clear and easy language to inform participants about the deceptive elements of the studies. Moreover, participants had the opportunity to contact the corresponding author should they have any questions.” (pp. 9-10)

Regarding the data sets, we apologize for this – we had mistakenly assumed that the data were readily available to all without any restrictions. We have now updated the settings in the respective OSF project and added the following sentence to the Method section of Study 1: 

“Data of all studies as well as the application for ethical approval (in German) and the ethics statement (in German) are available from the Open Science Framework (https://osf.io/kyxvf).” (p. 10)

5. P. 15: “ In line with our predictions, the effect of pro-diversity beliefs was smallest in the detrimental diverse condition”: This is true in absolute terms but you did not perform the statistical tests necessary to back up this claim. Other example, p. 16: “the magnitude of the positive effect of pro-diversity beliefs on all dependent variables was smaller when participants’ collaboration in diverse groups resulted in detrimental group performance”. In sum, please refrain from interpreting differences in absolute scores.

Thank you for this important comment. We now refrain from interpreting differences in absolute scores throughout the whole paper. We do, however, refer to a graphical inspection of conditional effect in some parts of the manuscript, as we feel this helps the reader comprehend some of the effects obtained.

6. P. 20: “Correlations between political orientation and prejudice (r = .533, p < .001) as well as social distance (r = .464, p < .001) were large. Hence, we included political orientation as a covariate in the respective analyses”: Is the mere fact of having high correlations sufficient to include political orientation as a covariate? I guess there must be some theoretical reason for doing this? Providing a strong justification for this is particularly important given that it changed the results.

Thank you for this comment. In fact, political orientation has been shown to influence attitudes towards ethnic outgroups and diversity in numerous studies (e.g., Jost, 2017, Pol Psych; Stewart, Gulzaib, & Morris, 2019; Front. Psychol). We, hence, considered this variable as a potential covariate while planning the studies. Following the recommendations by Wang et al. (2017, JESP), we included political orientation as a covariate in Studies 2 and 3 because it was highly correlated with the dependent variable. 

We now describe our approach on p. XYZ. The respective paragraph for Study 2 reads: 

“Political orientation was measured prior to the manipulation as a potential covariate [21]. In accordance with Wang, Sparks, Gonzales, Hess, & Ledgerwood [22], we had planned to include political orientation in our models only in case of high correlations with the dependent variables. Since the correlations between political orientation and dependent variables in Study 1, were only small to moderate however (general attitudes: r = -.368, p < .001; warmth: r = -.314, p = .001; competence: r = -.193, p = .044), we refrained from including political orientation as a covariate.” (p. 16)

We acknowledge that this approach might be controversial. We are, however, confident that we provide a maximum of transparency by reporting the results with and without political orientation for all studies. Nevertheless, we offer to consistently report the results without the covariate in the main text if the editor or the reviewers feel that would be a better approach. 

7. P. 24: “Moreover, we excluded an additional 71 participants who reported having studied abroad”: Was this pre-registered? If so, why did you decide to also run an analysis that included the data of these participants (see p. 27)?

Thank you for this valid question. In fact, in the preregistration we state:

“We will run and report analyses with and without participants with migration background and participants with own experiences with international study programs. However, we will focus on the reduced data set (i.e., dataset without respective participants).”

Since this is what we had stated in our pre-registration we decided to report the analyses of the data without exclusion of participants in the supplemental material. 

8. P. 28: “moderation of type of pro-diversity beliefs (i.e., instrumentality-based vs. justice-based pro-diversity beliefs) on prejudice by instrumentality”: difficult to understand. If possible, please clarify.

Thanking for pointing this out. We changed the respective paragraph. It now reads:

“We found a marginally significant interaction between type of pro-diversity beliefs (i.e., instrumentality-based vs. justice-based pro-diversity beliefs) and instrumentality on prejudice” (p. 47)

9. When submitting your revision, we need you to address these additional requirements. Please ensure that your manuscript meets PLOS ONE's style requirements, including those for file naming. The PLOS ONE style templates can be found at http://www.journals.plos.org/plosone/s/file?id=wjVg/PLOSOne_formatting_sample_main_body.pdf and http://www.journals.plos.org/plosone/s/file?id=ba62/PLOSOne_formatting_sample_title_authors_affiliations.pdf

Please include captions for your Supporting Information files at the end of your manuscript, and update any in-text citations to match accordingly. Please see our Supporting Information guidelines for more information: http://journals.plos.org/plosone/s/supporting-information.

Thank you for bringing up these points. We now adapted our manuscript to the respective style templates and added a caption for Supporting Information. We are sorry that our original manuscript had not met PLOS ONE’s style requirements from the outset. 

Comments by Reviewer #1: 

1. Overall framing: I think your point would be even clearer when you refer to it as a backlash effect from instrumentality-based pro-diversity beliefs, and would also explicitly call it like that in the title. That way your main point is clearer.

Thank you for this advice. Indeed, we ourselves have been thinking about this issue for some time also. Ultimately, however, we consciously decided against using the term backlash because it implies an adverse – often normative and/or motivational – reaction to a stimulus. For now, we can only speculate about the (motivational) processes underlying a potential moderation effect of instrumentality-based pro-diversity beliefs and actual instrumentality on outgroup attitudes. Moreover, as we point out throughout the paper, we do not necessarily expect negative instrumentality to lead to a reversed effect of instrumentality-based pro-diversity beliefs (it could also lead to a mitigating effect of positive effects of instrumentality-based pro-diversity beliefs).

2. I like and understand the positioning of the paper in the diversity beliefs literature, but one of the main questions I have is whether your findings are really due to experiences with diversity, or due to experiences with ethnic outgroup members. This is difficult/impossible to disentangle in your studies, but it does require discussing the literature on the contact hypothesis (e.g., Allport, 1954; Tropp & Pettigrew, 2005) and either providing an integrated perspective and argument, or discussing it as an alternative explanation for your findings. I guess the latter option would be easier, but the first could be more exciting and may actually not be too difficult given that the general idea of such an integration would be that positive [negative] experiences with a group of diverse members would positively [negatively] shape attitudes towards those members.

Thank you for this very useful recommendation. In the previous version of the manuscript, we had briefly referred to intergroup contact theory when discussing possible mechanisms of the observed effects. However, we agree that further discussion of this is important and to highlight that based on our studies we cannot rule out that experiencing (non-)instrumentality of diverse groups was interpreted as positive/negative intergroup contact experiences with ethnic outgroup members. We therefore added the following paragraph to the General Discussion: 

“On a related note, one could criticize that the present studies are unable to disentangle processes related to the instrumentality of diverse groups from more basic processes related to experiences with ethnic outgroup members. In other words, an alternative explanation for our findings could be that instrumentality-based pro-diversity beliefs interact with retroactive evaluations of interactions as positive and negative intergroup contact [32]. To understand this more fully further research is needed that addresses the interplay between instrumentality and more general valence of intergroup contact.” (p. 52) 

We, however, also want to stress that a negative group outcome does not necessarily mean that interactions partners perceived the interaction as negative. In fact, we believe that post-hoc perceptions of the valence of intergroup contact within diverse groups are more relevant here than the actual feelings encountered during the interactions.

3. Your intro and theory sections are well-written, but the comparison with pro-morality beliefs came too late for me. In the organizational literature, instrumentality-based pro-diversity beliefs are usually contrasted with instrumentality-based pro-similarity beliefs, so I was also surprised that you used a different comparison category. So it would be helpful if you introduce and explain morality-based pro-diversity beliefs earlier on as the comparison category.

Thank you for this valuable comment and for raising our awareness of this problem. We now refer to other forms of pro-diversity beliefs at the very beginning of the paper (i.e., at the end of the introduction section). The respective paragraph reads: 

“In order to test our assumptions, we measure instrumentality-based pro-diversity beliefs in Study 1 and experimentally contrast instrumentality-based pro-diversity beliefs with other, instrumentality-independent forms of pro-diversity beliefs (i.e. morality-based and justice-based pro-diversity beliefs) in Studies 2-4.” (p. 5)

Moreover, on pp. 7-8 we now state that “pro-diversity beliefs that do not presuppose instrumentality of diverse groups and are based on moral or justice-based considerations should be unaffected by the instrumentality of the instrumentality of diversity”. 

Finally, to clarify the difference between instrumentality-based and other pro-diversity beliefs we no longer use the term pro-diversity beliefs as a synonym for instrumentality-based pro-diversity beliefs – that is we now use the term instrumentality-based pro-diversity beliefs whenever referring to pro-diversity beliefs that rely on the business case argument. 

4. On p. 13, you discuss your operationalization of pro-diversity beliefs (IV) and of attitudes towards immigrants (DV ) in one paragraph. Please put them in different paragraphs to make it easier for the reader to understand.

Thank you very much for this suggestion. We have now put both descriptions in different paragraphs. 

5. In study 2, you excluded participants when they spent less than 10 seconds on both pages explaining the context, but the average time spent on those pages was much higher. So isn't 10 seconds still way too short and shouldn't the inclusion criteria be stricter (e.g., spending at least 25 seconds in order to have read it properly)?

Thank you for bringing up this issue. First of all, we wish to acknowledge that we initially found it difficult to set an inclusion criterion when it comes to the time spent on the pages comprising the manipulation texts, as it is difficult to gauge the exact amount time needed to read and comprehend the texts given. We decided to use 10 seconds as a cut-of criterion in Studies 2 and 3 because multiple (non-standardized) test runs indicated that it would be impossible to at least read the texts when spending less than 10 seconds on the respective pages. In other words, we believe that the 10-seconds-criterion can hence be regarded as an indicator for (not) clicking-through the pages. Participants that spent less than 10 seconds most certainly did not read the texts at all, yet participants that spent more than 10 seconds on the pages did read at least parts of the texts.

We thus considered 10 seconds as a conservative way to address this issue. Moreover, since we had set this criterion prior to data analyses (see preregistration of Study 3) we feel it would be arbitrary to adjust it now. And, the results concerning the manipulation checks in Studies 2 to 4 indicate that despite the relatively mild exclusion criterion our manipulations were successful.

6. To me, the design of study 2 came across as very obvious given that you measured participants' prejudice directly after the manipulation. It is also weird that the distractor items only came after measuring the DV, which means that there was little use of the distractor items for the study. So did you correctly write down the order of the study here, and if so, can you explain why you did it in this order and how you can be sure that participants were not aware of what your aims were in this study? And if it was too obvious, perhaps that also explains why you didn't find an interaction effect, given that the link between the manipulation and the DV was too strong? Either way, I think it would be good to critically consider whether the second study is of sufficient quality.

Thank you for this very valid point. First of all, we apologize for not having included sufficient detail on this in the previous version. We did not in fact present all of the distractor items after measuring the DVs, but had presented some distractor items prior to the measurement of the dependent variables while the other distractor items were presented on the same page as the dependent variables. 

However, we do now discuss potential effects of demand characteristics and its consequences. The respective paragraph reads: 

“although none of the participants indicated in their open feedback that they guessed the purpose of the study one could criticize that demand characteristics in Study 2 could have driven the effects. In fact, an inflated relationship between the instrumentality manipulation and the dependent variables could have impeded an interaction effect of pro-diversity beliefs.” (p. 31)

Finally and most importantly, in response to this very valid criticism we now included a replication of Study 2 that used a less obvious design, to mitigate the potential effects of demand characteristics on the results (as delineated above). 

7. In study 3, the reliability of your prejudice measure is below the standard threshold of a = .70. That does not necessarily constitute a problem, but at the very least you should point it out. Does the removal of an item yield a better score?

Thank you for this comment. We now state clearly that the scale’s reliability falls below the critical threshold of ɑ = .70. Moreover, we refer to reliabilities with and without exclusion of one item (see also our response to Reviewer 2’s comment 6).

8. In terms of interpretation, on p. 28 at the bottom you suggest that "presenting participants with instrumentality-based pro-diversity views surrounding exchange students increases negative attitudes towards this group when the presence of exchange students at universities was portrayed as having detrimental consequences". I understand that interpretation based on the comparison with the situation when the consequences were instrumental, but I wonder if that's the correct comparison. Shouldn't the comparison be with detrimental consequences under justice-based pro-diversity beliefs? In case instrumental pro-diversity beliefs even under detrimental consequences of diversity do not yield more negative attitudes towards outgroup members, then overall there is no problem with instrumental pro-diversity beliefs. However, in case instrumental pro-diversity beliefs do yield more negative attitudes under detrimental consequences compared to morality- or justice-based pro-diversity beliefs, then it really becomes questionable if instrumental pro-diversity beliefs are desirable. In other places (e.g., p. 30, the paragraph starting with "summarizing") your interpretations of your findings go into a similar direction, where I thus wonder if they shouldn't focus more on whether or not instrumentality-based pro-diversity beliefs yield more negative attitudes compared to morality-based pro-diversity beliefs.

Thank you for raising this important issue. We now report and discuss the comparison between the instrumentality-based pro-diversity beliefs detrimental and the morality/justice-based pro-diversity beliefs detrimental condition for Studies 2-4 (see also our response to Editor’s comment #2.

9. On a related point, you already mention in study 2 that you were somewhat surprised by the overall negative attitudes of those in a morality-based pro-diversity beliefs condition, and in general your findings do seem to indicate that when diversity is instrumental, instrumentality-based pro-diversity beliefs yield much more positive attitudes compared to morality-based pro-diversity beliefs. So you could also consider whether the actual story of the paper isn't that instrumental pro-diversity beliefs yield more positive/less negative attitudes when diversity also turns out to be instrumental - especially if it doesn't matter much whether people hold instrumentality- or morality-based pro-diversity beliefs when diversity is detrimental. Interestingly, such a conclusion would yield radically different implications. So I am looking forward to your reply.

Thank you for this important comment. In fact, after inclusion of an additional study, the pattern became somewhat less clear than before. Consequently, on the basis of the pattern of results obtained across all studies we do not believe that instrumentality-based pro-diversity beliefs yield more positive effects than morality- or justice-based pro-diversity beliefs when diversity is instrumental. We now include a discussion of this in the paper also – for example on p. 32:

“Besides a somewhat predictable main effect of instrumentality on the dependent variables (F(1, 298) = 8.25, p = .004, �2 = .027; social distance: F(1, 301) = 7.34, p = .007, �2 = .024), we found a negative main effect of morality-based pro-diversity beliefs (compared to instrumentality-based pro-diversity beliefs) on attitudes towards refugees prejudice (F(1, 298) = 5.11, p = .025, �2 = .017; social distance: F(1, 301) = 3.60, p = .059, �2 = .012) – an effect we had not expected. In fact, we had expected that activating moral reasons to support refugees by, for example, referring to human rights, should lead to more favorable attitudes towards refugees – regardless of the actual instrumentality [26]. What we might have overlooked here though is that the political and societal climate in Germany during the time of data collection was characterized by a polarization of political positions [27]. It is thus possible that emphasizing participants’ moral obligation to support refugees in the morality-based pro-diversity beliefs condition might have led participants to show reactance [28].”

Comments by Reviewer #2: 

1. While I have not recalculated the power analysis, I am little bit surprised that a sample of 100 participants (approx. 25 per cell) in Study 1 is sufficient for a moderation analysis. In light of the rather small sample, the authors might emphasize the marginal significant interactions too much (S1 warmth and competence), which are discussed as theoretical informative not only in Study 1 but also in Study 3 (even though I appreciated that the authors discuss this in the limitation section).

The power analyses (type of power analysis: F-Test, R2-increase) built on the effect size of �R² = .119 found in the pretest (other parameters: ɑ = .05; 1-� = .80; no. of predictors: 3) and reveals a total sample size of 96. However, we appreciate your comment and agree that the sample is rather small (and effects are quite weak). We therefore have now toned down the interpretations of results throughout the paper. 

2. I further wonder why the authors choose instrumental-diverse and not the neutral condition as baseline.

Thank you for this important question. We clearly could have done a better job in explaining why we chose to use the instrumental diverse condition as a baseline condition. We, hence, rewrote the respective paragraph (see response to Editor’s question #1). In principle, we believe that because previously found prejudice-reducing effects of instrumentality-based pro-diversity beliefs occurred because diversity was perceived as instrumental it makes sense to use the instrumental diverse condition as a reference group.

3. Study 1: The authors state that the items were assessed on a 5-point-Likert scale, yet the mean for general attitudes is reported as 6.31 (Table 2). Please correct this inconsistency. Please further discuss the overall very positive attitudes toward immigrants

This must be a misunderstanding. The first sentence of the Measures section of Study 1 states that “Unless otherwise indicated, all items were answered on 5-point-scales ranging from 1 = do not agree at all to 5 = totally agree”. Later, when describing the attitudes measures we describe that “Attitudes towards immigrants were measured with one feeling thermometer item (‘In general, how would you evaluate immigrants?’; scaling from 1 = very negative to 10 = very positive”.

Attitudes towards immigrants were thus not overly positive in this study (i.e., general attitudes: 6.21 (1.89) on a scale from 1 to 10; warmth: 3.31 (0.68) on a scale from 1 to 5; competence: 3.23 (0.73) on a scale from 1 to 5). Because the article is already quite long we refrained from discussing the means. However, if the editor/reviewers deem this to be an important issue we are more than happy to add more details on this to the Discussion of Study 1. 

4. Study 2: Please report the time frame of data collection

Thank you for this comment. While we had already included this information in the previous version of the manuscript, we must confess that it was somewhat hidden in the middle of the Methods section and thus easy to miss. We now clearly state the time frame of data collection for each study at the beginning of the Participants section. 

5. Please discuss the unexpected finding that morality-based diversity beliefs lead to detrimental effects in greater details.

Thank you for this comment. We now extended the discussion of this effect (see p. 32). Moreover, we refer to this discussion in the section in which we first mention this particular result.

6. Please report by how much the two deleted items (S1 and S3) affected the reliability of the scale

Thank you for raising our awareness that this is relevant information for readers. We now included the reliability of scales prior to exclusion of the respective items in the Measures sections of Studies 1 and 2.

---

## [Decision Letter · Decision Letter 1]

1 May 2020

PONE-D-19-24506R1

When good for business is not good enough: Effects of pro-diversity beliefs and instrumentality of diversity on intergroup attitudes

PLOS ONE

Dear Dr. Kauff,

Thank you for sending us the revised version of your paper. The reviewers and I are grateful for the many changes that you made, especially the inclusion of the new pre-registered study. As a result, the paper has improved considerably. The reviewers still have some relatively minor comments that they would like you to address in a second revision of the paper. Hence, I invite you to revise the paper once more.

We would appreciate receiving your revised manuscript by Jun 15 2020 11:59PM. To enhance the reproducibility of your results, we recommend that if applicable you deposit your laboratory protocols in protocols.io, where a protocol can be assigned its own identifier (DOI) such that it can be cited independently in the future. For instructions see: http://journals.plos.org/plosone/s/submission-guidelines#loc-laboratory-protocols

We look forward to receiving your revised manuscript.

Kind regards,

Jan De Houwer

Academic Editor

PLOS ONE

Reviewers' comments:

Reviewer's Responses to Questions

**Comments to the Author**

1. If the authors have adequately addressed your comments raised in a previous round of review and you feel that this manuscript is now acceptable for publication, you may indicate that here to bypass the “Comments to the Author” section, enter your conflict of interest statement in the “Confidential to Editor” section, and submit your "Accept" recommendation.

Reviewer #1: (No Response)

Reviewer #2: All comments have been addressed

2. Is the manuscript technically sound, and do the data support the conclusions?

Reviewer #1: Yes

Reviewer #2: Partly

3. Has the statistical analysis been performed appropriately and rigorously? 

Reviewer #1: Yes

Reviewer #2: Yes

4. Have the authors made all data underlying the findings in their manuscript fully available?

Reviewer #1: Yes

Reviewer #2: Yes

5. Is the manuscript presented in an intelligible fashion and written in standard English?

Reviewer #1: Yes

Reviewer #2: Yes

6. Review Comments to the Author

Reviewer #1: Thank you very much for addressing the comments and suggestions in a diligent way. I really like your new, third study, appreciate the nuances in the design and that it was preregistered, and overall think that your paper provides an interesting contribution to the literature. I do have a number of suggestions to further bolster the message of your paper.

1. I still think that your findings do not only speak about the potential danger of instrumentality-based pro-diversity beliefs, but also provide some insights into how people look at what you call morality-based pro-diversity beliefs (but see next comment). Studies 1 and 2 indicate that there is a main effect of type of pro-diversity beliefs on prejudice, such that instrumentality-based pro-diversity beliefs are associated with more positive/less negative attitudes. Whereas I think that your concerns about instrumentality-based pro-diversity beliefs are valid, I think your paper also raises questions about the value of morality-based pro-diversity beliefs. In general, I wonder if people aren't wary of moralism - being told what they should or shouldn't do or endorse. In particular in study 2 that may be true. In your discussion you also raise the question if alternative rationales for pro-diversity beliefs wouldn't be better, and by being more explicit about the potential negative effects of morality-based pro-diversity beliefs, you would bolster the need for an alternative pro-diversity rationale. You now only very briefly raise the functionality of morality-based pro-diversity beliefs on p. 53, but think it would be much better if you dedicate more attention to it.

2. Consistency in terminology is very important, especially when some terms are similar. You now sometimes use morality-based and justice-based pro-diversity beliefs exchangably. My suggestion is to label those justice-based pro-diversity beliefs, given that instrumentality-based pro-diversity beliefs technically are also grounded in a moral paradigm (i.e. utilitarianism), so suggesting that only justice-based pro-diversity beliefs are moral would be inaccurate. I know that in the literature the "moral" case for diversity is often mentioned as the opposite of the business case and that it thus would be more consistent to label it morality-based pro-diversity beliefs, but labeling it justice-based would be more accurate.

3. Regarding the design of study 3: Given the pool where respondents came from, I assume that not all respondents were German. Do you think that may have influenced the findings, given that they are asked to reflect on the situation in Germany? Is there any way to control of non-Germans gave different responses than Germans?

4. It is sometimes a bit confusing that you focus on instrumentality-based pro-diversity beliefs and look at whether diversity is instrumental. The overlap in terminology is logical but also confusing. Is it an idea to focus on whether diversity is "beneficial" instead of instrumental to enhance the readability?

5. On page 4 and throughout the introduction and the theory, you do raise the question if instrumentality-based pro-diversity beliefs may lead to negative attitudes in case diversity is detrimental, but do not explain the potential reasons why that may be the case. You only briefly speculate about that in the discussion and indicate that more research is needed for that. I can imagine that you don't want to mention those reasons in the theory section because you're not testing the underlying mechanisms, but now it comes across as if you are studying something just for the sake of studying it (see also page 3 where the main rationale for this study is that it hasn't been studied before). I think you can provide a main argument of why instrumentality-based pro-diversity beliefs may backfire in your intro and theory without needing to actually study that potential underlying mechanis, and that doing so would help to make readers understand why your study is important.

6. I appreciate that you now mention "moral or justice-based considerations" as the comparison category before the methods, but just providing those terms without explaining it still doesn't add much. So it would be helpful if you insert a paragraph in which you explain what justice-based pro-diversity beliefs entail. By explaining the/an alternative of instrumentality-based pro-diversity beilefs, readers will also be better able to understand what is meant with instrumentality-based pro-diversity beliefs and what potential problems with it may be.

Some textual points:

p. 5 last word: heterogeneous  don't you mean homogeneous?

p. 7, 5th line from the bottom: better replace "should" with "may" because it's not a normative argument that you're making here.

p. 41, 5th line: weak instead of week

p. 50, after (Study 1) remove thus

p. 50, after In Study 3 remove however

p. 52, asset-driven views is a term you haven't used before. Probably best to just stick to instrumentality-based views or something related that you've mentioned before

Reviewer #2: I believe the authors responded reasonably well to my comments. I appreciate that the authors toned down their language throughout the manuscript, that they uploaded their materials one OSF, and that they carried out an additional pre-registered study. I have only some minor comments.

1) In the ethical statement, the authors registered a total of 6 studies. In the present manuscript the authors report, however, only 4 studies one of which (Study 3) has been conducted 2019/2020. Please state in the paper why you did not include the other studies.

2) In study 2, contact experiences have been assessed as well. I am surprised that the authors decide to not include previous contact experiences as a moderator or a co-variate. Particularly, because the authors refer to contact theory in their discussion and state that “… in times of heated political debates strategies involving the promotion of diversity on moral grounds can also backfire and may even lead individuals to more strongly oppose diversity [27]. A potentially more promising alternative could involve the promotion of more inclusive social identity processes.” I think it’s reasonable to assume that individuals reporting more contact experiences with refugees might have a more inclusive identity.

3) Study 4: Was the data part of a larger project? Point 1 is missing in the pre-registration and the Figure suggests that categorization has been assessed as well. Please state whether the same data is used elsewhere.

7. PLOS authors have the option to publish the peer review history of their article (what does this mean?). If published, this will include your full peer review and any attached files.

Reviewer #1: Yes: Hans van Dijk

Reviewer #2: No

---

## [Author Response · Author response to Decision Letter 1]

19 May 2020

Dear Professor De Houwer, 

Thank you for your encouraging comments on the revised version of our manuscript. We would also like to thank both reviewers for their constructive suggestions. We feel that they have helped us to further improve the clarity of the paper. 

We detail our response to each of the points raised by the reviewers below. 

Thank you again for your valuable suggestions and for giving us the chance to revise our manuscript. 

Sincerely,

[the authors]

Reviewer #1:

1. Thank you very much for addressing the comments and suggestions in a diligent way. I really like your new, third study, appreciate the nuances in the design and that it was preregistered, and overall think that your paper provides an interesting contribution to the literature. I do have a number of suggestions to further bolster the message of your paper.

Thanks a lot for this favorable evaluation of our revised manuscript! 

2. I still think that your findings do not only speak about the potential danger of instrumentality-based pro-diversity beliefs, but also provide some insights into how people look at what you call morality-based pro-diversity beliefs (but see next comment). Studies 1 and 2 indicate that there is a main effect of type of pro-diversity beliefs on prejudice, such that instrumentality-based pro-diversity beliefs are associated with more positive/less negative attitudes. Whereas I think that your concerns about instrumentality-based pro-diversity beliefs are valid, I think your paper also raises questions about the value of morality-based pro-diversity beliefs. In general, I wonder if people aren't wary of moralism - being told what they should or shouldn't do or endorse. In particular in study 2 that may be true. In your discussion you also raise the question if alternative rationales for pro-diversity beliefs wouldn't be better, and by being more explicit about the potential negative effects of morality-based pro-diversity beliefs, you would bolster the need for an alternative pro-diversity rationale. You now only very briefly raise the functionality of morality-based pro-diversity beliefs on p. 53, but think it would be much better if you dedicate more attention to it.

Thank you for this comment. We agree that a potential rejection of moral/justice-based arguments when it comes to diversity is an interesting and important avenue for (future) research. To that end we think the results of Study 2 with regard to morality-based pro-diversity beliefs are worth discussing in more detail. We therefore added some additional arguments in the respective section. Having said this, we were worried of over-interpreting this particular finding of Study 2. Although we find a negative main effect of justice-based pro-diversity beliefs on prejudice (but not on social distance) in Study 2, we do not find such an effect in Studies 3 and 4. It is for this reason that we decided to not explicitly state that our paper raises questions about the value of morality-based pro-diversity beliefs.

“Critics on the business case for diversity would probably agree with this idea and advise building a case for diversity on the grounds of equality-based and moral arguments [1]. However, in Study 2 we found a prejudice-increasing effect of morality-based pro-diversity beliefs (compared to instrumentality-based pro-diversity beliefs). We did not however find the same effect in Studies 3 and 4, although morality- and justice-based pro-diversity beliefs did also not reduce prejudice in these studies. Accordingly, it is questionable to assume that moral arguments in favor of diversity can be considered as a panacea when it comes to support for diversity and ethnic outgroups. At least in times of heated political debates, strategies involving the promotion of diversity on moral grounds can backfire and may even lead individuals to more strongly oppose diversity [27].”(p. 54) 

3. Consistency in terminology is very important, especially when some terms are similar. You now sometimes use morality-based and justice-based pro-diversity beliefs exchangably. My suggestion is to label those justice-based pro-diversity beliefs, given that instrumentality-based pro-diversity beliefs technically are also grounded in a moral paradigm (i.e. utilitarianism), so suggesting that only justice-based pro-diversity beliefs are moral would be inaccurate. I know that in the literature the "moral" case for diversity is often mentioned as the opposite of the business case and that it thus would be more consistent to label it morality-based pro-diversity beliefs, but labeling it justice-based would be more accurate.

Thank you for this important and helpful comment. We now labeled the non-instrumentality-based conditions justice-based condition in all studies. 

4. Regarding the design of study 3: Given the pool where respondents came from, I assume that not all respondents were German. Do you think that may have influenced the findings, given that they are asked to reflect on the situation in Germany?

Thank you for raising this question. In fact, all participants were German and lived in Germany. Clickworker as well as Prolific operate in Germany with German participants. MTurkers were only allowed to participate when they were native Germans and were currently living in Germany. We now describe the sampling strategy as follows:

“German participants were recruited via three crowdsourcing/online recruiting websites” (p. 35)

5. It is sometimes a bit confusing that you focus on instrumentality-based pro-diversity beliefs and look at whether diversity is instrumental. The overlap in terminology is logical but also confusing. Is it an idea to focus on whether diversity is "beneficial" instead of instrumental to enhance the readability?

Thank you for bringing up this issue. We are aware that the similar terminology might be a bit confusing at times, and indeed had ourselves reflected on this issue repeatedly. In the end, however, we decided to use the term “instrumental”, as we believe that it is important to underline the logical link between instrumentality-based pro-diversity beliefs and actual instrumentality by using related terms. However, we now refer whenever possible to actual instrumentality to emphasize the differences between expectations about diversity implied in instrumentality-based pro-diversity beliefs and the real or actual diversity encountered when interacting within diverse groups. 

6. On page 4 and throughout the introduction and the theory, you do raise the question if instrumentality-based pro-diversity beliefs may lead to negative attitudes in case diversity is detrimental, but do not explain the potential reasons why that may be the case. You only briefly speculate about that in the discussion and indicate that more research is needed for that. I can imagine that you don't want to mention those reasons in the theory section because you're not testing the underlying mechanisms, but now it comes across as if you are studying something just for the sake of studying it (see also page 3 where the main rationale for this study is that it hasn't been studied before). I think you can provide a main argument of why instrumentality-based pro-diversity beliefs may backfire in your intro and theory without needing to actually study that potential underlying mechanis, and that doing so would help to make readers understand why your study is important.

Thank you for bringing this issue up. We, now, refer to the mechanisms in the theory. The respective paragraph reads: 

“Hence, instrumentality-based pro-diversity beliefs may only lead to an improvement of intergroup attitudes as long as diversity is actually perceived as instrumental. Indeed, based on the reasoning by Noon [6] and others [1], instrumentality-based pro-diversity beliefs may legitimize the devaluation of outgroup members, that is it may even negatively impact intergroup attitudes if, counter to expectations, diversity ends up hindering group success. Besides being a legitimizing process, it might also be that instrumentality-based pro-diversity beliefs intensify subgroup-categorization which could also facilitate the devaluation of outgroup members – especially when expectations are not met [12].” (p. 8)

7. I appreciate that you now mention "moral or justice-based considerations" as the comparison category before the methods, but just providing those terms without explaining it still doesn't add much. So it would be helpful if you insert a paragraph in which you explain what justice-based pro-diversity beliefs entail. By explaining the/an alternative of instrumentality-based pro-diversity beilefs, readers will also be better able to understand what is meant with instrumentality-based pro-diversity beliefs and what potential problems with it may be.

Thank you for raising this point again. Re-reading our manuscript we agree that we had not done enough to illustrate the idea of justice/morality-based pro-diversity beliefs. We have now attended to this, and in the newly revised version of the manuscript we therefore now state:

“Justice-based pro-diversity beliefs do not build on an instrumentality-based business case for diversity. They do not imply the idea that diversity should be valued because of its anticipated instrumentality. Rather they refer to groups’ and individuals’ moral obligations to reduce inequality and injustice between social groups. In other words, in a justice-based pro-diversity beliefs framework diversity is valued because of individuals’ duty to work against inequality and to support others that are in need [1, 6].” (p. 8) 

8. Some textual points: p. 5 last word: heterogeneous  don't you mean homogeneous?; p. 7, 5th line from the bottom: better replace "should" with "may" because it's not a normative argument that you're making here; p. 41, 5th line: weak instead of week; p. 50, after (Study 1) remove thus; p. 50, after In Study 3 remove however; p. 52, asset-driven views is a term you haven't used before. Probably best to just stick to instrumentality-based views or something related that you've mentioned before

Thank you very much for your careful proof-reading while reviewing our manuscript, and for these very helpful suggestions and corrections. We have now revised the respective sentences in the revised manuscript.

Reviewer #2:

1. I believe the authors responded reasonably well to my comments. I appreciate that the authors toned down their language throughout the manuscript, that they uploaded their materials one OSF, and that they carried out an additional pre-registered study. 

Thank you for this positive evaluation of the revised version of our manuscript!

2. In the ethical statement, the authors registered a total of 6 studies. In the present manuscript the authors report, however, only 4 studies one of which (Study 3) has been conducted 2019/2020. Please state in the paper why you did not include the other studies.

Thank you for this important comment. As a matter of fact, we did not run all six studies (and also slightly modified the ones we did implement). We, now, added this information:

“Please note, that in contrast to the planned research project outlined in the application for ethical approval, we only realized three instead of six studies and slightly modified some studies with regard to measurement or circumstantial research design matters. Moreover, we ran an additional replication study. Our procedure with regard to research ethics was not altered, however.” (p. 10)

3. In study 2, contact experiences have been assessed as well. I am surprised that the authors decide to not include previous contact experiences as a moderator or a co-variate. Particularly, because the authors refer to contact theory in their discussion and state that “… in times of heated political debates strategies involving the promotion of diversity on moral grounds can also backfire and may even lead individuals to more strongly oppose diversity [27]. A potentially more promising alternative could involve the promotion of more inclusive social identity processes.” I think it’s reasonable to assume that individuals reporting more contact experiences with refugees might have a more inclusive identity.

Thank you for your suggestion to include intergroup contact as an additional covariate in Study 2. Contact was not significantly associated with the dependent variables. We, hence, decided to refrain from including it. However, we added the following sentence: 

“As mentioned above, we also measured intergroup contact with refugees prior to the manipulation. Because contact was not correlated with neither prejudice (r = .016, p = .766) nor social distance (r = .086, p = .133) we refrained from including it as an additional covariate.” (p. 28)

4. Study 4: Was the data part of a larger project? Point 1 is missing in the pre-registration and the Figure suggests that categorization has been assessed as well. Please state whether the same data is used elsewhere. 

Thank you raising our awareness of this potential misunderstanding. As we point out on p. 43, we had to discard items measuring categorization after a non-editable version of the preregistration was created. As a result, the actual study design unfortunately slightly differs from the preregistered design. 

To avoid ambiguity regarding the use of data, we added a sentence stating that data of all studies were exclusively used for the present paper. 

“Data of the four studies presented in this paper are exclusively used for these studies, that is the data reported here have not been, and will not be, used for other projects.” (p. 10)

---

## [Editor Report · Decision Letter 2]

21 May 2020

When good for business is not good enough: Effects of pro-diversity beliefs and instrumentality of diversity on intergroup attitudes

PONE-D-19-24506R2

Dear Dr. Kauff,

Thanks for revising the paper a second time. I am pleased to inform you that your manuscript has been judged scientifically suitable for publication and will be formally accepted for publication once it complies with all outstanding technical requirements. I have noticed, however, that the data files that you put on the OSF have the extension ".sav" (SPSS files). SPSS is a licenced software that many colleagues do not have access to. Hence, I strongly encourage you to also make available your data in an open source format (e.g., ".csv").

With kind regards,

Jan De Houwer

Academic Editor

PLOS ONE

---

## [Editor Report · Acceptance letter]

22 May 2020

PONE-D-19-24506R2 

When good for business is not good enough: Effects of pro-diversity beliefs and instrumentality of diversity on intergroup attitudes 

Dear Dr. Kauff:

I am pleased to inform you that your manuscript has been deemed suitable for publication in PLOS ONE. Congratulations! Your manuscript is now with our production department. 

With kind regards,

on behalf of

Dr. Jan De Houwer 

Academic Editor

PLOS ONE